

# Advancements in heuristic task scheduling for IoT applications in fog-cloud computing: challenges and prospects

Deafallah Alsadie

Department of Computer Science and Artificial Intelligence, College of Computing, Umm Al-Qura University, Makkah, Makkah Almukaramah, Saudi Arabia

## ABSTRACT

Fog computing has emerged as a prospective paradigm to address the computational requirements of IoT applications, extending the capabilities of cloud computing to the network edge. Task scheduling is pivotal in enhancing energy efficiency, optimizing resource utilization and ensuring the timely execution of tasks within fog computing environments. This article presents a comprehensive review of the advancements in task scheduling methodologies for fog computing systems, covering priority-based, greedy heuristics, metaheuristics, learning-based, hybrid heuristics, and nature-inspired heuristic approaches. Through a systematic analysis of relevant literature, we highlight the strengths and limitations of each approach and identify key challenges facing fog computing task scheduling, including dynamic environments, heterogeneity, scalability, resource constraints, security concerns, and algorithm transparency. Furthermore, we propose future research directions to address these challenges, including the integration of machine learning techniques for real-time adaptation, leveraging federated learning for collaborative scheduling, developing resource-aware and energy-efficient algorithms, incorporating security-aware techniques, and advancing explainable AI methodologies. By addressing these challenges and pursuing these research directions, we aim to facilitate the development of more robust, adaptable, and efficient task-scheduling solutions for fog computing environments, ultimately fostering trust, security, and sustainability in fog computing systems and facilitating their widespread adoption across diverse applications and domains.

# INTRODUCTION

Fog computing stands as a transformative breakthrough, reshaping the landscape of traditional cloud computing by extending its reach to the network's edge (*Fahad et al., 2022*; *Madhura, Elizabeth & Uthariaraj, 2021*). This innovation enables real-time processing, data analytics, and application deployment in close proximity to data sources, diverging significantly from centralized cloud architectures. Particularly tailored for Internet of Things (IoT) environments, fog computing accommodates the diverse array of interconnected devices generating data at the network's periphery, each with its

Corresponding author
Deafallah Alsadie,
dbsadie@uqu.edu.sa

computational capabilities and requirements (*Azizi et al., 2022*; *Abd Elaziz, Abualigah & Attiya, 2021*).

In the realm of fog computing, effective task scheduling emerges as a paramount challenge. Task scheduling involves the reasonable allocation of computing resources to tasks, aiming to optimize performance metrics such as makespan, energy consumption, and resource utilization (*Jamil et al., 2022*). However, achieving optimal task scheduling in fog computing environments proves inherently intricate due to the dynamic nature of the network, the diverse array of heterogeneous computing resources available, and the stringent constraints imposed by edge devices (*Bansal, Aggarwal & Aggarwal, 2022*; *Subbaraj & Thiyagarajan, 2021*; *Kaur, Kumar & Kumar, 2021*).

In light of these challenges, this article embarks on a comprehensive review of task scheduling methodologies tailored for fog computing systems. Through meticulous analysis and evaluation, a spectrum of heuristic approaches is scrutinized, encompassing priority-based strategies, greedy heuristics, metaheuristic algorithms, learning-based approaches, hybrid heuristics, and nature-inspired methodologies. This review critically assesses the strengths, limitations, and practical applications of each approach within the context of fog computing environments.

The primary objective of this review is to provide researchers, practitioners, and stakeholders in the fog computing domain with a thorough understanding of the state-of-the-art task scheduling methodologies. By synthesizing insights from existing literature and delineating key challenges and prospective research trajectories, this article aims to propel the field of fog computing task scheduling forward. Ultimately, this collective effort seeks to catalyze the development of more resilient, adaptable, and efficient solutions tailored to meet the demands of real-world applications.

This study makes significant contributions in the following aspects:

1. **Comprehensive review:** The article offers a comprehensive review of task scheduling methodologies for fog computing systems, encompassing various heuristic approaches, including priority-based, greedy heuristics, metaheuristics, learning-based, hybrid heuristics, and nature-inspired heuristics.

2. **Systematic analysis:** Through a systematic analysis of relevant literature, the article evaluates the strengths and limitations of each task scheduling approach, offering insights into their effectiveness in optimizing resource allocation, improving energy efficiency, and ensuring timely task execution.

3. **Identification of challenges:** The article identifies key challenges facing fog computing task scheduling, such as dynamic environments, heterogeneity, scalability issues, resource constraints, security concerns, and algorithm transparency, providing a clear understanding of the obstacles to overcome in the field.

4. **Proposal of future research directions:** By proposing future research directions, including the integration of machine learning approaches, leveraging federated learning, developing resource-aware and energy-efficient algorithms, incorporating security-aware approaches, and advancing explainable AI methodologies, the article guides researchers

towards addressing the identified challenges and advancing the field of fog computing task scheduling.

5. **Facilitation of widespread adoption:** Through its insights and recommendations, the article aims to facilitate the development of more robust, adaptable, and efficient task scheduling solutions for fog computing environments, ultimately fostering trust, security, and sustainability in fog computing systems and promoting their widespread adoption across diverse applications and domains.

The subsequent sections of this article follow a structured outline: section titled "Methodology" elaborates on the comprehensive methodology employed for this review. The article then advances to section titled "Taxonomy of heuristic approaches for task scheduling in fog computing", where a detailed taxonomy is presented, classifying heuristic methods for further analysis in the study. Section titled "Heuristic approaches for task scheduling" provides a comprehensive overview of various heuristic methods as per the defined taxonomy, critically analyzing them based on Approach, Performance & Optimization, Implementation & Evaluation, Performance Metrics, Strengths, and Limitations. In section titled "Open challenges and future directions", challenges in efficient task scheduling are underscored, followed by an exploration of future directions and emerging trends in "Heuristic-based task scheduling methods". This section discusses potential advancements and critical research areas. Section titled "Conclusion" provides a summary of the main findings and underscores the significance of tackling the challenges associated with heuristic-based task scheduling in fog computing environments.

# METHODOLOGY

This systematic literature review on heuristic-based task scheduling in fog computing follows the guidelines established in the Preferred Reporting Items for Systematic Reviews and Meta-Analyses (PRISMA) (*Page et al., 2021*). The subsequent sections detail the specific steps and methodology employed in this review.

## Inclusion criteria for reviewed studies

This comprehensive review focuses on diverse studies employing heuristic-based task scheduling in fog computing. The inclusive search criteria encompassed various factors, including terms such as heuristics, optimization, and nature-inspired methods for task scheduling in fog computing. The selected studies were required to meet criteria related to task scheduling, energy optimization, and resource management. Moreover, stringent eligibility criteria for report characteristics, such as English language publication, classification as a scientific article or review, and a publication year between 2019–2024, were meticulously applied. The review followed the PRISMA protocol, ensuring a systematic and standardized selection of studies. Electronic databases like IEEE Xplore, ScienceDirect, and SpringerLink were utilized for the search, along with consideration of highly cited articles from ACM, MDPI, De Gruyter, Hindawi, and Wiley, focusing on task scheduling in fog computing environments, specifically heuristic-based methods.

## Search strategy

The selection of search keywords adhered to the defined review framework. Primary concepts for the search included "task scheduling," "heuristic," "metaheuristic," "learning-based," "fog computing," "nature-inspired,", "genetic algorithm," "simulated annealing," "reinforcement learning," and "tabu search" with logical operators (AND/OR) between each keyword. Exclusion criteria were applied based on language, expressly limited to English.

## Record selection procedure

Upon acquiring records from database searches and manual exploration, they were imported into the JabRef reference management system for comprehensive review. The authors manually compared titles and authors to identify and eliminate duplicate records. Subsequently, the reviewers meticulously examined each article retrieved from the search, assessed its eligibility, and collaboratively reached a majority consensus to finalize the selection of studies included in the review.

## Data collection process

Reviewers actively engaged in evaluating and analyzing the studies included in the review. Data collection was facilitated through a Google spreadsheet, where information from the selected studies was systematically compiled. The resulting document emerged as an advanced matrix, offering a thorough insight into the cutting-edge developments within the field. Each row within the matrix corresponded to an individual study, while the columns were dedicated to distinct data elements earmarked for analysis.

## Data items

The collaborative spreadsheet's columns were thoughtfully designated to align with the specific outcomes for which data was being sought. The defined columns encompassed year, title, authors, general approach, performance & optimization, implementation & evaluation, performance metrics, strengths, and limitations. This meticulous structuring allowed for a systematic and comprehensive collection of relevant data points from the reviewed studies. Articles published from 2019 to March 02, 2024, were scrutinized to comprehend the trends and advancements in this study area. This systematic review identified and included 102 articles that met the predetermined inclusion criteria.

There is a noticeable interest in the field as per the upward trend apparent from the increasing number of publications in various indexed journal databases since 2019, as depicted in Fig. 1. Figure 2 provides an overview of the distribution of selected articles by the publisher, focusing on both journal and conference contributions. The qualified articles in our sample set employ diverse heuristic-based methods, including priority-based, greedy, metaheuristics, learning-based, hybrid and nature-inspired heuristic methods as depicted in Fig. 3.

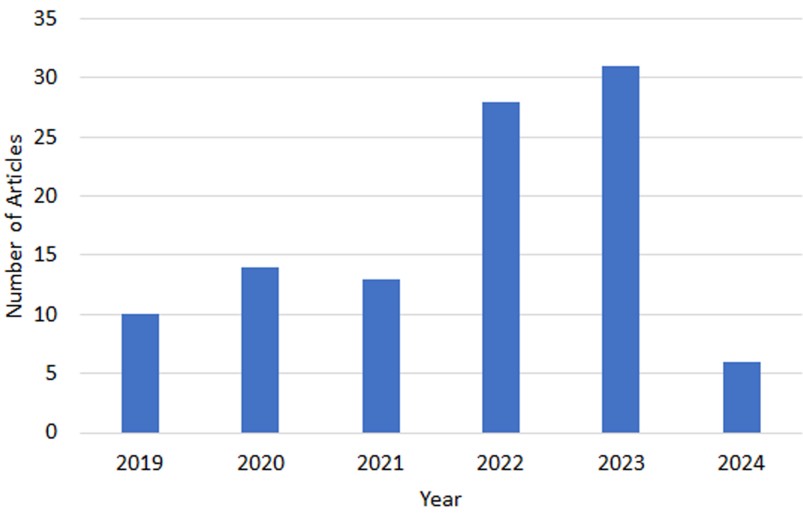

**Figure 1 Trends of heuristic-based task scheduling studies.**

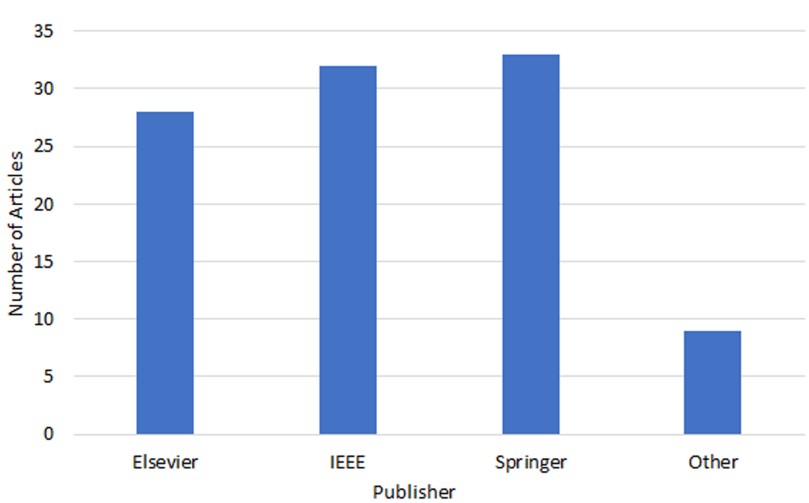

**Figure 2 Publisher distribution of heuristic-based task scheduling studies.**

# TAXONOMY OF HEURISTIC APPROACHES FOR TASK SCHEDULING IN FOG COMPUTING

In the realm of fog computing environments, effective task scheduling stands as a pivotal factor in optimizing resource usage and elevating the performance of IoT applications. This section introduces a taxonomy outlining heuristic methods for task scheduling in fog computing, organizing them according to their fundamental principles and attributes. Refer to Fig. 4 for a visual representation of this taxonomy.

1. Priority-based heuristics: Priority-based heuristics prioritize task execution based on predefined criteria (*Fahad et al., 2022*; *Tang et al., 2023*). Static Priority Scheduling assigns fixed priorities to tasks, typically determined by factors such as deadlines, importance, or resource requirements. In contrast, Dynamic Priority Scheduling adjusts
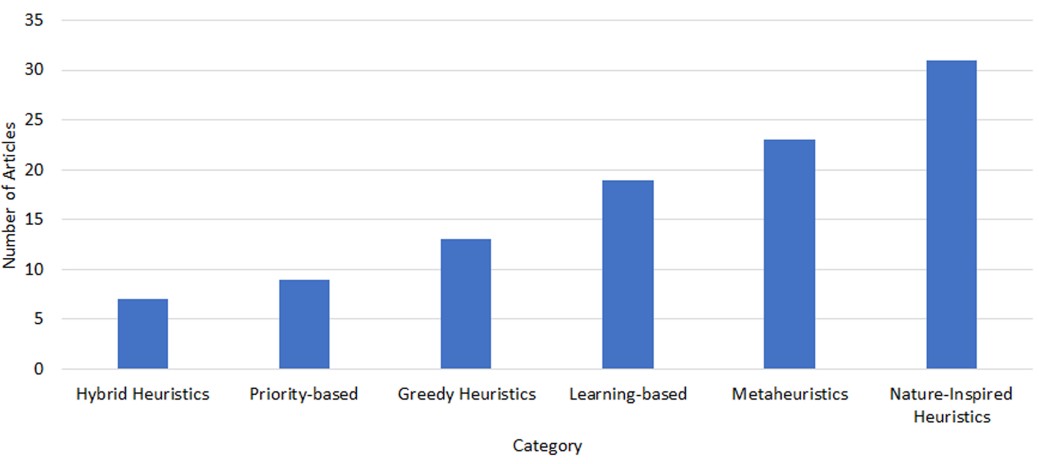

**Figure 3 Heuristic method distribution of task scheduling studies.**

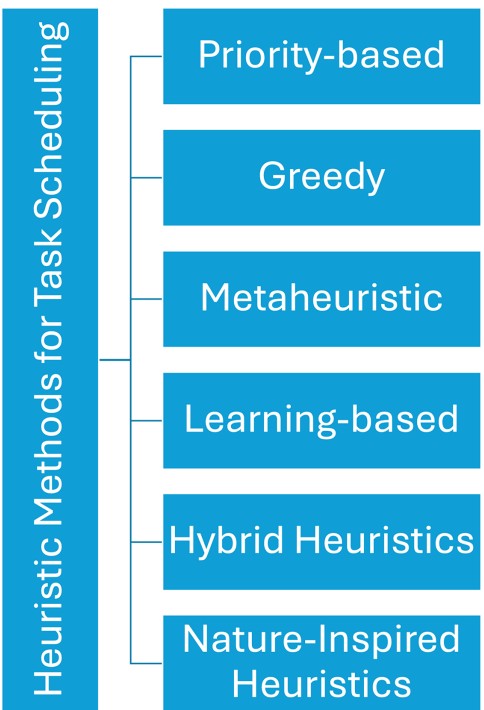

**Figure 4 Taxonomy of heuristic approaches for task scheduling in fog computing.**

task priorities dynamically during runtime in response to real-time system conditions, workload characteristics, or user-defined policies (*Shi et al., 2020*).

2. Greedy heuristics: Greedy heuristics make locally optimal decisions at each step with the aim of achieving a globally optimal solution (*Azizi et al., 2022*). Earliest deadline first (EDF) schedules tasks based on their earliest deadlines, prioritizing those with imminent deadlines to minimize lateness. Shortest processing time (SPT) selects tasks with the shortest estimated processing time, aiming to minimize overall completion time and

improve system throughput. Minimum remaining processing time (MRPT) prioritizes tasks based on their remaining processing time, with shorter tasks given precedence to expedite completion.

3. Metaheuristic approaches: Metaheuristic approaches are high-level strategies that guide the search for optimal solutions in a solution space (*Wu et al., 2022*; *Keshavarznejad, Rezvani & Adabi, 2021*). Genetic algorithms (GA) employ genetic operators to evolve a population of candidate solutions towards an optimal task schedule. Particle swarm optimization (PSO) mimics the collective behavior of a swarm of particles to iteratively explore the solution space and converge towards an optimal task schedule. Ant colony optimization (ACO) draws inspiration from the foraging behavior of ants, utilizing pheromone trails and heuristic information to navigate towards an optimal task schedule on a global scale. Simulated annealing (SA) simulates the gradual cooling of a material to find the global optimum by accepting probabilistic changes in the solution (*Dev et al., 2022*).

4. Learning-based heuristics: Learning-based heuristics leverage machine learning approaches to discover optimal task scheduling policies (*Wang et al., 2024*). Reinforcement learning (RL) utilizes trial-and-error learning to discover optimal task scheduling policies through interactions with the environment and feedback on task completion. Q-Learning learns an optimal action-selection strategy by iteratively updating a Q-table based on rewards obtained from task scheduling decisions (*Gao et al., 2020*; *Yeganeh, Sangar & Azizi, 2023*). Deep Q-Networks (DQN) extend Q-learning by employing deep neural networks to approximate the Q-function, enabling more complex and scalable task scheduling policies.

5. Hybrid heuristics: Hybrid heuristics integrate multiple heuristic approaches to exploit their complementary strengths and improve solution quality (*Agarwal et al., 2023*; *Yadav, Tripathi & Sharma, 2022a*). This includes combinations of greedy and metaheuristic approaches, as well as the fusion of learning-based and metaheuristic approaches (*Leena, Divya & Lilian, 2020*).

6. Nature-inspired heuristics: Nature-inspired heuristics draw inspiration from natural phenomena to develop efficient task-scheduling strategies (*Mishra et al., 2021*; *Usman et al., 2019*). This encompasses biologically inspired algorithms, such as genetic evolution and swarm intelligence, as well as physics-based heuristics derived from principles in physics to optimize task allocation and resource utilization in fog environments.

The taxonomy of heuristic methods provides a comprehensive framework for understanding the diverse approaches to task scheduling in fog computing, each offering unique advantages and applications in optimizing the performance of IoT applications as presented in Table 1.

## HEURISTIC APPROACHES FOR TASK SCHEDULING

Task scheduling in fog computing relies heavily on heuristic approaches to allocate computational resources efficiently and meet the dynamic demands of IoT applications.

**Table 1 Taxonomy of heuristic methods.**

| Category | Method | Advantages | Disadvantages | Suitability |
|---|---|---|---|---|
| Priority-based | Static priority | Simple, fast, predictable | Ignores dynamic changes, suboptimal results | Static, non-critical tasks |
| | Dynamic priority | Adapts to real-time, flexible | Requires accurate priorities, complex implementation | Dynamic, heterogeneous environments |
| Greedy | EDF | Deadline-aware, low overhead | Starves long tasks, sensitive to deadlines | Real-time, deadline-critical applications |
| | SPT | Improves throughput, fast execution | Starves long tasks, ignores resources | Non-critical, independent tasks |
| | MRPT | Prioritizes near completion, reduces waiting times | Task preemption overhead, favors short tasks | Bursty workloads, quick task completion |
| Metaheuristic | GA | Robust search, global optimization | High complexity, tuning required | Large-scale, complex tasks |
| | PSO | Efficient exploration, fast convergence | Sensitive to parameters, local optima | Dynamic, moderate-sized tasks |
| | ACO | Flexible, diverse environments | Parameter tuning, slow convergence | Multi-objective optimization, heterogeneous resources |
| | SA | Escapes local optima, complex problems | Slow convergence, careful cooling schedule | Highly constrained, critical tasks |
| Learning-based | RL | Adapts to dynamics, learns policies | High training overhead, large datasets, exploration-exploitation | Dynamic, data-driven applications |
| | Q-Learning | Simple implementation, no model | Slow convergence, large state-action space | Small-scale tasks, moderate complexity |
| | DQN | Handles complex state-action spaces, faster learning | Large datasets, computationally expensive | Large-scale, data-rich environments |
| Hybrid | Greedy+ Metaheuristic | Combines strengths, improves quality | Increased complexity, parameter tuning | Complex, dynamic, diverse objectives |
| | Learning + Metaheuristic | Learns from data, enhances exploration-exploitation | Highly complex, specialized expertise | Large-scale, data-driven, evolving requirements |
| Nature-inspired | Biologically-inspired | Efficient, flexible, adaptable | Domain-specific knowledge, complex implementation | Diverse applications, innovation potential |
| | Physics-based | Energy-efficient, distributed, scalable | Specific optimization problems, not readily applicable | Resource-constrained, energy-aware scheduling |

Heuristic approaches encompass various categories, each presenting distinct strategies for optimizing task scheduling. These categories are elucidated in the subsequent subsections and consolidated in Table 2.

## Priority-based heuristics

Priority-based heuristics provide a straightforward method for task scheduling in fog computing environments, where tasks are prioritized based on predefined criteria such as deadlines, importance, or resource requirements (*Sharma & Thangaraj, 2024*; *Choudhari, Moh & Moh, 2018*). Static priority scheduling assigns fixed priorities to tasks, offering simplicity but lacking adaptability to dynamic environments. In contrast, dynamic priority scheduling adjusts priorities based on real-time conditions, offering flexibility at the cost of

**Table 2 Category-wise distribution of studies utilizing heuristic approaches.**

| Category | Studies |
|---|---|
| Priority-based heuristics | *Fahad et al. (2022)*, *Madhura, Elizabeth & Uthariaraj (2021)*, *Movahedi, Defude & Hosseininia (2021)*, *Hoseiny et al. (2021)*, *Choudhari, Moh & Moh (2018)*, *Jamil et al. (2022)*, *Bansal, Aggarwal & Aggarwal (2022)*, *Subbaraj & Thiyagarajan (2021)*, *Kaur, Kumar & Kumar (2021)*, *Wu et al. (2022)* |
| Greedy heuristics | *Azizi et al. (2022)*, *Zavieh et al. (2023)*, *Tang et al. (2023)*, *Haja, Vass & Toka (2019)*, *Liu, Lin & Buyya (2022)*, *Chen et al. (2023)*, *Adewojo & Bass (2023)* |
| Metaheuristics | *Hosseinioun et al. (2020)*, *Hosseini, Nickray & Ghanbari (2022)*, *Apat et al. (2019)*, *Jalilvand Aghdam Bonab & Shaghaghi Kandovan (2022)*, *Huang & Wang (2020)*, *Yadav, Tripathi & Sharma (2022a)*, *Abdel-Basset et al. (2023)*, *Abd Elaziz, Abualigah & Attiya (2021)*, *Yadav, Tripathi & Sharma (2022b)*, *Abd Elaziz & Attiya (2021)*, *Saif et al. (2023)*, *Abohamama, El-Ghamry & Hamouda, 2022*, *Nguyen et al. (2020)*, *Mousavi et al. (2022)*, *Salehnia et al. (2023)*, *Nazeri, Soltanaghaei & Khorsand (2024)*, *Memari et al. (2022)*, *Hussain & Begh (2022)*, *Dev et al. (2022)*, *Javaheri et al. (2022)*, *Keshavarznejad, Rezvani & Adabi (2021)*, *Hajam & Sofi (2023)*, *Hussein & Mousa (2020)*, *Yadav, Tripathi & Sharma (2023)*, *Abdel-Basset et al. (2020)*, *Kishor & Chakarbarty (2022)*, *Khaledian et al. (2024)*, *Khiat, Haddadi & Bahnes (2024)*, *Ahmadabadi, Mood & Souri (2023)* |
| Learning-based heuristics | *Fellir et al. (2020)*, *Wang et al. (2024)*, *Liu et al. (2016)*, *Ibrahim & Askar (2023)*, *Gao et al. (2020)*, *Siyadatzadeh et al. (2023)*, *Gazori, Rahbari & Nickray (2020)*, *Guevara et al. (2022)*, *Tahmasebi-Pouya, Sarram & Mostafavi (2023)*, *Raju & Mothku (2023)*, *Farhat, Sami & Mourad (2020)*, *Shi et al. (2020)*, *Jain & Kumar (2023)*, *Mishra et al. (2023)*, *Yeganeh, Sangar & Azizi (2023)*, *Fahimullah et al. (2023)*, *Devarajan et al. (2023)*, *Zheng et al. (2022a)*, *Sellami et al. (2022)*, *Jayanetti, Halgamuge & Buyya (2022)*, *Wu et al. (2018)*, *Sun, Lin & Xu (2018)*, *Ali et al. (2020)*, *Ramezani Shahidani et al. (2023)*, *Baek et al. (2019)*, *Jie et al. (2021)*, *Xiong et al. (2020)*, *Wang et al. (2019a)*, *Huang et al. (2019)*, *Chen et al. (2018)*, *Zheng et al. (2022b)*, *Zhao, Li & He (2023)*, *Liao et al. (2023)*, *Sethi & Pal (2023)*, *Wang et al. (2019b)* |
| Hybrid heuristics | *Agarwal et al. (2023)*, *Alqahtani, Amoon & Nasr (2021)*, *Aron & Abraham (2022)*, *Gupta & Singh (2023)*, *Leena, Divya & Lilian (2020)*, *Mtshali et al. (2019)*, *Huang, Zhang & Wang (2023)* |
| Nature-inspired heuristics | *Usman et al. (2019)*, *Chhabra et al. (2022)*, *Shao, Fu & Wang (2023)*, *Abd Elaziz, Abualigah & Attiya (2021)*, *Dabiri, Azizi & Abdollahpouri (2022)*, *Khan et al. (2019)*, *Khaledian et al. (2023)*, *Saif et al. (2022)*, *Matrouk & Matrouk (2023)*, *Mishra et al. (2021)*, *Kaushik & Al-Raweshidy (2022)*, *Dubey & Sharma (2023)*, *Abdel-Basset et al. (2021)*, *Nematollahi, Ghaffari & Mirzaei (2023)*, *Jiang et al. (2024)* |

complexity. These heuristics find applications in real-time systems and resource-constrained environments but face challenges such as oversimplification, sensitivity to priority settings, and the potential for task starvation. While effective for specific scenarios, their limitations underscore the need for exploring more sophisticated approaches like learning-based or hybrid heuristics to address the complexities of fog computing environments.

*Fahad et al. (2022)* introduced a preemptive task scheduling strategy tailored for fog computing environments, known as multi-queue priority (MQP) scheduling. This approach aims to address the challenge of task starvation among less critical applications while ensuring balanced task allocation for both latency-sensitive and less latency-sensitive tasks. Tasks are categorized into short and long based on their processing duration, with dynamically updated preemption time slots. Through an intelligent traffic management case study, the effectiveness of the MQP algorithm in reducing service latencies for long tasks was demonstrated. Simulation outcomes showcased significant reductions in latency compared to alternative scheduling algorithms. The proposed approach targets response time reduction for both latency-sensitive and less latency-sensitive tasks, mitigating the starvation issue for less latency-sensitive tasks by maintaining separate task queues and dynamically adjusting preemption time slots. Simulation findings revealed the efficient

task allocation capabilities of the MQP algorithm, resulting in reduced service latencies for long tasks. Across all experimental configurations, an average percentage reduction in latency of 22.68% and 38.45% was achieved compared to First Come-First Serve and shortest job first algorithms, respectively.

*Madhura, Elizabeth & Uthariaraj (2021)* introduced an innovative task scheduling algorithm tailored specifically for fog computing environments. The algorithm, consisting of three distinct phases namely level sorting, task prioritization, and task selection, aimed at minimizing both makespan and computation costs. Recognizing the pivotal role of efficient task scheduling in achieving high-performance program execution, especially in scenarios involving tasks represented as directed acyclic graphs (DAGs) with precedence constraints, the proposed algorithm strategically allocated tasks based on the computation cost of the node and the task's execution finishing time. Extensive experimentation with randomly generated graphs and real-world data showcased the algorithm's superior performance compared to existing approaches such as predicting earliest finish time, heterogeneous earliest finish time algorithm, minimal optimistic processing time, and SDBBATS. Performance metrics including average scheduling length ratio, speedup, and makespan underscored the algorithm's efficacy in enhancing task scheduling efficiency in fog computing environments.

*Movahedi, Defude & Hosseininia (2021)* addressed the task scheduling challenge within fog computing environments by introducing a novel method termed OppoCWOA, which harnesses the Whale Optimization Algorithm (WOA). This approach integrates opposition-based learning and chaos theory to augment the efficacy of WOA in optimizing task scheduling. The study exemplified the application of this approach in an intelligent city scenario, elucidating a hierarchical fog-based architecture for task scheduling. Furthermore, the task scheduling conundrum was formulated as a multi-objective optimization problem using integer linear programming (ILP). Through extensive experimentation, the authors compared the performance of OppoCWOA against established meta-heuristic optimization algorithms like PSO, ABC, and GA. The findings underscored the superior efficiency of OppoCWOA, particularly in optimizing time and energy consumption metrics.

*Hoseiny et al. (2021)* addressed the challenges inherent in executing Internet of Things (IoT) tasks within fog-cloud computing environments, which offer low latency but encounter resource constraints. Their article introduced a scheduling algorithm named PGA, designed to optimize overall computation time, energy consumption, and the percentage of tasks meeting deadlines. Leveraging a hybrid approach and genetic algorithm, the PGA algorithm accounts for task requirements and the heterogeneous characteristics of fog and cloud nodes when assigning tasks to computing nodes. Through simulations, the study demonstrated the algorithm's superiority over existing strategies. As the IoT ecosystem continues to expand rapidly, efficient processing and networking resources become increasingly vital, with cloud computing playing a crucial role in meeting the demands of IoT applications sensitive to latency.

*Choudhari, Moh & Moh (2018)* proposed a task scheduling algorithm within the fog layer, employing priority levels to accommodate the growing number of IoT devices while

enhancing performance and reducing costs. Their work meticulously described the proposed architecture, queueing and priority models, along with the priority assignment module and task scheduling algorithms. Performance evaluations illustrated that, in comparison to existing algorithms, the proposed approach reduced overall response time and significantly diminished total costs. The article underscored fog computing's role as a model that introduces a virtualized layer between end-users and cloud data centers, aiming to mitigate transmission and processing delays for IoT systems. It highlighted the importance of this research in advancing fog computing technology and its potential applicability across diverse domains.

*Jamil et al. (2022)* conducted a thorough and systematic comparative analysis, examining various scheduling algorithms, optimization metrics, and evaluation methodologies within the context of fog computing and the Internet of Everything (IoE). The primary objectives of their survey encompassed several key aspects: firstly, to provide an overview of fog computing and IoE paradigms; secondly, to delineate pertinent optimization metrics tailored to these environments; thirdly, to classify and compare existing scheduling algorithms, supplemented with illustrative examples; fourthly, to rationalize the efficacy of these algorithms and derive insights from the survey findings; and fifthly, to discuss unresolved challenges and outline prospective research avenues in fog computing and IoE domains. The study addressed a spectrum of issues pertaining to resource allocation and task scheduling across fog computing, IoE, cloud computing, and related paradigms. Moreover, the article delved into the significance of simulation and modeling tools, underscoring the utility of advanced methodologies such as deep reinforcement learning in navigating the intricate landscape of task scheduling. In addition to offering a comprehensive review, the article furnished a plethora of references and recommendations for further exploration in this burgeoning field.

*Bansal, Aggarwal & Aggarwal (2022)* explored diverse scheduling approaches utilized in fog computing, categorizing them into static, dynamic, heuristic, and hybrid approaches. They revealed that 17% of researchers employed static methods, while 23% utilized dynamic approaches, with heuristic approaches being the most prevalent at 47%, followed by hybrid strategies at 13%. Researchers primarily focused on QoS parameters such as response time (19%), cost and energy consumption (18%), and makespan (16%).

*Subbaraj & Thiyagarajan (2021)* delved into the challenges posed by the increasing number of IoT devices, emphasizing the issues with transmitting real-time sensor data to cloud data centers due to security risks and high costs. Introducing fog computing as a solution, which distributed computing resources closer to the devices, the focus shifted to resource allocation and task scheduling in fog computing, accounting for the heterogeneity of fog devices. The proposed work aimed to utilize multi-criteria decision-making approaches for module mapping in fog environments to meet application performance requirements. It outlined the implementation stages and simulation results, highlighting the heterogeneity of fog devices. The conclusion underscored the significance of the proposed MCDM-based scheduling algorithm in heterogeneous fog environments. This study introduced a novel model for task-resource mapping focused on optimizing performance in fog computing. This model considered multiple performance metrics

including MIPS, RAM, storage, latency, bandwidth, trust, and cost. To evaluate fog device performance and allocate tasks effectively, two multi-criteria decision-making techniques, namely Analytic Hierarchy Process (AHP) and Technique for Order Preference by Similarity to Ideal Solution (TOPSIS), were employed. Simulation outcomes demonstrated the superior performance of the proposed method compared to alternative scheduling algorithms in fog environments. Notably, the proposed approach took into consideration performance, security, and cost metrics when making scheduling decisions.

*Kaur, Kumar & Kumar (2021)* conducted an investigation into the intricacies of task scheduling within Fog computing, shedding light on the obstacles associated with accommodating dynamic user demands amid resource limitations. Their study underscored the challenges arising from the heterogeneous nature of Fog nodes, alongside constraints pertaining to task deadlines, cost considerations, and energy constraints. In response to these complexities, the article proposed a comprehensive taxonomy of research challenges and pinpointed notable gaps in existing methodologies. Additionally, it examined prevalent solutions, conducted a meta-analysis of quality of service parameters, and scrutinized the tools employed for implementing task scheduling algorithms in Fog environments. By undertaking this systematic review, the authors aimed to furnish researchers with valuable insights for identifying specific research gaps and delineating future avenues to enhance scheduling efficacy in Fog computing landscapes.

*Wu et al. (2022)* proposed an innovative solution named Improved Parallel Genetic Algorithm for IoT Service Placement (IPGA-SPP) to tackle the IoT service placement problem (SPP) within fog computing. To mitigate the risk of genetic algorithms converging to local optima, IPGA-SPP was parallelized with shared memory and elitist operators, thereby enhancing its performance. The approach addressed load balancing by considering resource distribution and prioritizing service execution to minimize latency. Moreover, IPGA-SPP treated SPP as a multi-objective problem, maintaining a set of Pareto solutions to optimize service latency, cost, resource utilization, and service time simultaneously. A notable aspect of this scheme was the integration of a two-way trust management mechanism to ensure trustworthiness between clients and service providers, a crucial yet often overlooked aspect in fog computing solutions. Through simulation in a synthetic fog environment, IPGA-SPP demonstrated an average performance improvement of 8.4% compared to existing methods like CSA-FSPP, GA-PSO, EGA, and WOA-FSP. This approach offers a comprehensive solution that considers latency, cost, and trust aspects, effectively addressing the challenges of IoT service placement while prioritizing Quality of Service (QoS) and security.

Table 3 provides a comparative summary of studies focusing on priority-based heuristic approaches for task scheduling.

## Greedy heuristics

In task scheduling, where the efficient allocation of tasks to resources is crucial, greedy heuristics provide a rapid and straightforward approach, prioritizing tasks based on specific criteria without guaranteeing the best solution (*Azizi et al., 2022*). Standard methods like EDF, SPT, and MRPT exemplify this approach, aiming to find satisfactory

**Table 3  Comparison of priority-based heuristic studies for task scheduling.**

| Study | General approach | Performance & optimization | Implementation & evaluation | Performance metrics | Strengths | Limitations |
|---|---|---|---|---|---|---|
| *Fahad et al. (2022)* | MQP based preemptive task scheduling approach | Balanced task allocation for latency-sensitive and less latency-sensitive applications, addressing task starvation for less important tasks | Smart traffic management case study; Simulation results compared to other scheduling algorithms | Service latencies for long tasks; Average percentage reduction in latency | Reduced service latencies; Separate task queues for each category | Limited mention of scalability and real-world deployment |
| *Madhura, Elizabeth & Uthariaraj (2021)* | List-based task scheduling algorithm | Minimizing makespan and computation cost | Experimentation with random graphs and real-world data | Average scheduling length ratio, speedup, makespan | Superior performance compared to existing algorithms | Lack of scalability analysis |
| *Movahedi, Defude & Hosseininia (2021)* | OppoCWOA utilizing the WOA | Integration of opposition-based learning and chaos theory to enhance WOA performance | Comparative experimentation with other meta-heuristic optimization algorithms | Time and energy consumption optimization | Improved performance compared to other algorithms | Limited discussion on scalability and real-world applicability |
| *Hoseiny et al. (2021)* | PGA algorithm considering task requirements and fog/cloud node heterogeneity | Optimization of computation time, energy consumption, and percentage of deadline satisfied tasks | Simulations demonstrating algorithm superiority over existing strategies | Overall computation time, energy consumption, and percentage of satisfied tasks | Efficient resource allocation; Utilization of hybrid approach and genetic algorithm | Lack of extensive scalability analysis |
| *Choudhari, Moh & Moh (2018)* | Task scheduling algorithm based on priority levels for fog layer | Supporting increasing number of IoT devices, improving performance, and reducing costs | Performance evaluation showcasing reduction in overall response time and total cost | Overall response time and total cost | Reduced response time and decreased total cost | Limited scalability discussion |
| *Jamil et al. (2022)* | Comparative study exploring various scheduling algorithms, optimization metrics, and evaluation tools | Reviewing fog computing and IoE paradigms, classification and comparison of scheduling algorithms, discussion of open issues | N/A | N/A | Comprehensive review providing valuable insights and recommendations | No specific implementation or evaluation |
| *Bansal, Aggarwal & Aggarwal (2022)* | Exploration of scheduling approaches in fog computing categorized into static, dynamic, heuristic, and hybrid approaches | Examination of scheduling approaches, prevalence analysis, discussion of tools and open issues | N/A | N/A | Thorough categorization and analysis | Lack of specific implementation details |

(Continued)

| Study | General approach | Performance & optimization | Implementation & evaluation | Performance metrics | Strengths | Limitations |
|---|---|---|---|---|---|---|
| *Subbaraj & Thiyagarajan (2021)* | Proposed MCDM-based scheduling algorithm for module mapping in fog environments | Utilization of multi-criteria decision-making approaches for module mapping, considering Fog node heterogeneity | Simulation-based evaluation; Performance-oriented task-resource mapping | MIPS, RAM & storage, latency, bandwidth, trust, cost | Superior performance over other scheduling algorithms | Limited scalability analysis |
| *Kaur, Kumar & Kumar (2021)* | Exploration of challenges and gaps in task scheduling in Fog computing | Comprehensive taxonomy of research issues, meta-analysis on QoS parameters, review of scheduling tools | N/A | N/A | Identification of research problems and future directions | No specific implementation or evaluation |
| *Wu et al. (2022)* | Introduction of IPGA for IoT SPP in fog computing | Parallel configuration with shared memory and elitist operators, multi-objective approach, trust management mechanism | Simulation in synthetic fog environment | Service latency, cost, resource utilization, service time | Enhanced deployment process, latency-aware solution | Limited discussion on real-world applicability and scalability |

solutions quickly. While advantageous due to their simplicity, efficiency, and adaptability, greedy heuristics could fall short in guaranteeing optimality and could be sensitive to initial conditions, potentially resulting in suboptimal outcomes. Nevertheless, they excelled in real-time scheduling, large-scale problems, and more straightforward scenarios, offering valuable solutions despite their limitations. Awareness of their strengths and weaknesses was essential for effectively leveraging greedy heuristics in task-scheduling endeavors.

*Azizi et al. (2022)* developed a mathematical formulation for the task scheduling problem aimed at minimizing the overall energy consumption of fog nodes (FNs) while ensuring the fulfillment of Quality of Service (QoS) criteria for IoT tasks and minimizing deadline violations. They introduced two semi-greedy-based algorithms, namely priority-aware semi-greedy (PSG) and PSG with a multistart procedure (PSG-M), designed to efficiently allocate IoT tasks to FNs. Evaluation metrics encompassed the percentage of IoT tasks meeting their deadlines, total energy consumption, total deadline violation time, and system makespan. Results from experiments showcased that the proposed algorithms enhanced the percentage of tasks meeting deadlines by up to 1.35 times and reduced total deadline violation time by up to 97.6% compared to the next-best outcomes. Moreover, optimization of fog resource energy consumption and system makespan was achieved. The article underscores the complexities inherent in deploying fog computing resources for IoT

applications with real-time constraints, introduces novel algorithms to address these challenges, and outlines potential avenues for future research.

*Zavieh et al. (2023)* introduced a novel methodology termed the Fuzzy Inverse Markov Data Envelopment Analysis Process (FIMDEAP) to tackle the task scheduling and energy consumption challenges prevalent in cloud computing environments. By integrating the advantages of Fuzzy Inverse Data Envelopment Analysis (FIDEA) and Fuzzy Markov Decision Process (FMDP) techniques, this approach adeptly selected physical and virtual machines while operating under fuzzy conditions. Data representation utilized triangular fuzzy numbers, and problem-solving relied on the alpha-cut method. The authors presented a mathematical optimization model and provided a numerical example for clarification purposes. Furthermore, the performance of FIMDEAP was rigorously assessed through simulations conducted in a cloud environment. Results exhibited superior performance compared to existing methods like PSO+ACO and FBPSO+FBACO across critical metrics such as energy consumption, execution cost, response time, gain of cost, and makespan. This innovative method not only enhanced energy optimization but also improved response time and makespan, thereby promoting the adoption of environmentally sustainable practices in cloud networks. Future research avenues were suggested, including the exploration of additional fuzzy methods, integration of machine learning techniques, and incorporation of request forecasting to further refine resource allocation and task processing optimization strategies.

*Tang et al. (2023)* conducted a study on AI-driven IoT applications within a collaborative cloud-edge environment, leveraging container technology. They introduced a novel container-based task scheduling algorithm dubbed PGT, which integrates a priority-aware greedy strategy with the Technique for Order Preference by Similarity to Ideal Solution (TOPSIS) multi-criteria approach. This algorithm effectively manages containers across both cloud and edge servers within a unified platform, facilitating the deployment of IoT application services within these containers. Tasks are prioritized based on their deadline constraints, with higher precedence given to tasks with shorter deadlines. The proposed algorithm takes into account various performance indicators, including task response time, energy consumption, and task execution cost, to determine the optimal container for task execution. Through simulations conducted in a collaborative cloud-edge environment, the study demonstrated that the proposed scheduling approach surpasses four baseline algorithms in enhancing the Quality of Service (QoS) satisfaction rate, reducing energy consumption, minimizing penalty costs, and mitigating total violation time.

*Haja, Vass & Toka (2019)* outlined their efforts towards improving network-aware big data task scheduling in distributed systems. They proposed several resource orchestration algorithms designed to address challenges related to network resources in geographically distributed topologies, specifically focusing on reducing end-to-end latency and efficiently allocating network bandwidth. The heuristic algorithms they introduced demonstrated enhanced performance for big data applications compared to default methods. These solutions were implemented and evaluated within a simulation environment, showcasing the improved quality of big data application outcomes. On a related note, the text

introduced concepts like edge-cloud computing and mobile edge computing, which involved deploying computing resources closer to end devices to minimize network latency. These approaches, integrated with carrier networks, supported various 5G and beyond applications like Industry 4.0, Tactile Internet, remote driving, and extended reality, facilitating low-latency communication and task offloading from end devices to the distributed environment.

*Liu, Lin & Buyya (2022)* tackled the scheduling issue by framing it as a modified version of bin-packing and proposed an algorithm driven by heuristics to decrease inter-node communication. They introduced D-Storm, a prototype scheduler developed using the Apache Storm framework, which integrated a self-adaptive MAPE-K (Monitoring, Analysis, Planning, Execution, Knowledge) architecture. By conducting assessments with real-world applications like Twitter sentiment analysis, the researchers illustrated that D-Storm outperformed both the existing resource-aware scheduler and the default Storm scheduler. Notably, D-Storm achieved reductions in inter-node traffic and application latency while also realizing resource savings through task consolidation. This contribution enhances the management and scheduling of data streams in cloud and sensor networks, showcasing tangible enhancements in performance and resource utilization.

*Chen et al. (2023)* addressed the challenges of task offloading and resource scheduling in vehicular edge computing, proposing a Multi-Aerial Base Station Assisted Joint Computation Offload algorithm based on D3QN in Edge VANETs (MAJVD3). This algorithm utilized SDN Controllers to efficiently schedule resources and tackle issues such as latency, energy consumption, and QoS degradation. The proposed method underwent evaluation and comparison with baseline algorithms, showcasing enhancements in network utility, decreased task latency, and energy consumption. The article discussed related work, the system model, optimization problem formulation, the proposed algorithm, simulation results, and analysis. The authors declared no competing interests and acknowledged funding support. *Adewojo & Bass (2023)* proposed a novel load balancing algorithm aimed at efficiently distributing workload among virtual machines for three-tier web applications. The algorithm combines five carefully selected server metrics to achieve load distribution. Experimental evaluations were conducted on a private cloud utilizing OpenStack to compare the proposed algorithm's performance with a baseline algorithm and round-robin approach. Performance was assessed under scenarios involving simulated resource failures and flash crowds, with response times meticulously recorded. Results indicated a 12.5% improvement in average response times compared to the baseline algorithm and a 22.3% improvement compared to the round-robin algorithm during flash crowds. Additionally, average response times were enhanced by 20.7% compared to the baseline algorithm and 21.4% compared to the round-robin algorithm during resource failure situations. These experiments underscored the novel algorithm's resilience to fluctuating loads and resource failures, showcasing its effectiveness in dynamic scenarios. The text also acknowledged limitations and proposed avenues for future research to further enhance the algorithm's efficacy.

Table 4 provides a comparative summary of studies focusing on greedy heuristic approaches for task scheduling.

**Table 4 Comparison of greedy heuristic studies for task scheduling.**

| Study | General approach | Performance & optimization | Implementation & evaluation | Performance metrics | Strengths | Limitations |
|---|---|---|---|---|---|---|
| *Azizi et al. (2022)* | Mathematical formulation, semi-greedy algorithms | Minimization of energy consumption, meeting QoS requirements | Simulation experiments | Deadline compliance, energy consumption, makespan | Improvements in task deadline meeting, energy consumption | Limited scalability discussion |
| *Zavieh et al. (2023)* | FIMDEAP | Optimization of multiple metrics using fuzzy methods | Mathematical optimization model, cloud simulation | Energy consumption, execution cost, response time | Outperforms existing methods in key metrics | Requires exploration of additional fuzzy methods |
| *Tang et al. (2023)* | PGT container-based scheduling algorithm | Improvement of QoS, energy consumption | Simulation in cloud-edge environment | QoS satisfaction rate, energy consumption | Outperforms baseline algorithms | Real-world applicability may need further investigation |
| *Haja, Vass & Toka (2019)* | Resource orchestration algorithms | Reduction of latency, bandwidth optimization | Simulation experiments | End-to-end latency, bandwidth utilization | Enhanced performance compared to default methods | Limited discussion on real-world deployment |
| *Liu, Lin & Buyya (2022)* | D-Storm scheduler for data streams | Minimization of inter-node communication, resource savings | Evaluation with real-world applications | Inter-node traffic, application latency | Improvements in performance and resource utilization | Limited scalability discussion |
| *Chen et al. (2023)* | MAJVD3 algorithm for vehicular edge computing | Reduction of latency, energy consumption | Simulation experiments | Network utility, task latency, energy consumption | Improvements in network utility and latency | Real-world deployment considerations may be needed |
| *Adewojo & Bass (2023)* | Load balancing algorithm for web applications | Efficient workload distribution | Experimental characterization in private cloud | Average response time under varying scenarios | Resilience to fluctuating loads and failures | Limited scalability discussion |

## Metaheuristics

Metaheuristic approaches in task scheduling optimization provided adaptable and versatile solutions by heuristically exploring the search space (*Aron & Abraham, 2022*; *Gupta & Singh, 2023*). Examples include genetic algorithms, particle swarm optimization, simulated annealing, and ant colony optimization. These methods iteratively refined task assignments using feedback mechanisms to converge towards optimal or near-optimal solutions (*Khan et al., 2019*). Metaheuristics excelled in complex and dynamic IoT environments where traditional scheduling approaches might have been insufficient, handling diverse optimization objectives and adapting to changing conditions. However, they might have demanded substantial computational resources and parameter tuning, rendering them less suitable for real-time applications with stringent performance constraints. Various metaheuristic approaches used in task scheduling encompass simulated annealing (SA), tabu search (TS), particle swarm optimization (PSO), genetic algorithm (GA), harmony search (HS), differential evolution (DE), firefly algorithm (FA), bat algorithm (BA), cuckoo search (CS), and WOA.

*Hosseinioun et al. (2020)* proposed an energy-aware method by employing the Dynamic Voltage and Frequency Scaling (DVFS) technique to reduce energy consumption. Additionally, to construct valid task sequences, a hybrid approach combining the Invasive Weed Optimization and Culture (IWO-CA) evolutionary algorithm was utilized. The experimental results indicated that the proposed algorithm enhanced existing methods in terms of energy consumption. The article addressed the challenges encountered in cloud computing and underscored the role of fog computing in mitigating these challenges, particularly in minimizing latency and energy consumption. It emphasized the importance of task scheduling in distributed systems and introduced a hybrid IWO-CEA algorithm aimed at achieving energy-aware task scheduling in fog computing environments. Furthermore, the article highlighted the use of the DVFS technique to minimize energy consumption while maximizing resource utilization. Experimental results were presented to illustrate the effectiveness of the proposed algorithm. *Hosseini, Nickray & Ghanbari (2022)* presented a scheduling algorithm called PQFAHP, leveraging a combination of Priority Queue, Fuzzy logic, and Analytical Hierarchy Process (AHP). The PQFAHP algorithm was implemented to integrate diverse priorities and rank tasks according to multiple criteria. These criteria encompassed dynamic scheduling parameters such as completion time, energy consumption, RAM usage, and deadlines. Experimental outcomes underscored the efficacy of the proposed method in integrating multi-criteria for scheduling tasks, surpassing several standard algorithms in key performance metrics including waiting time, delay, service level, mean response time, and the number of scheduled tasks on the mobile fog computing (MFC) platform. The study highlighted significant advancements in fog computing scheduling, including notable reductions in average waiting time, delay, and energy consumption, alongside enhancements in service level. It emphasized the importance of addressing the challenges faced by IoT in MFC environments and proposed a comprehensive solution to optimize resource allocation, minimize execution costs, and enhance system performance. Additionally, the article provided a critical review of existing methodologies in the field, underscoring the need for more effective scheduling algorithms to meet the demands of mobile fog computing.

*Apat et al. (2019)* focused on mapping independent tasks to the fog layer, where the proposed algorithm demonstrated better performance compared to the cloud data center. The system resources considered included CPU, RAM, *etc.*, with task priorities determined based on deadlines. Additionally, the assumption was made that once a task was assigned to a specific node, it remained there until completion. The article introduced a three-layer architecture designed for efficient task scheduling, particularly in applications like healthcare within smart homes. The text also delved into the limitations of cloud computing, citing issues like connection interruptions and high latency, especially concerning IoT devices. It underscored the increasing significance of IoT devices in daily life while highlighting the mismatch between the centralized nature of cloud computing and the decentralized nature of IoT, which often led to latency issues impacting service quality.

*Jalilvand Aghdam Bonab & Shaghaghi Kandovan (2022)* presented a novel framework for QoS-aware resource allocation and mobile edge computing (MEC) in multi-access

heterogeneous networks, with the objective of maximizing overall system energy efficiency while ensuring user QoS requirements. The framework introduced a customized objective function tailored specifically for the multi-server MEC environment, taking into account computation and communication models to minimize task completion time and enhance energy efficiency within specified delay constraints. By integrating continuous carrier allocation, user association variables, and interference coordination into the objective function, the core optimization problem was reformulated as a mixed integer nonlinear programming (MINLP) task. Moreover, a carrier-matching algorithm was proposed to tackle constraints related to user data rate and transmission power, thereby optimizing the channel allocation strategy. Through extensive simulations, the proposed approach exhibited notable enhancements in energy efficiency and network throughput, particularly evident in multi-source scenarios.

*Huang & Wang (2020)* introduced a novel framework, referred to as GO, designed to tackle large-scale bilevel optimization problems (BOPs) efficiently. Divided into two main phases, GO first identified and categorized interactions between upper-level and lower-level variables into three subgroups. These subgroups dictated the optimization approach utilized in the subsequent phase. For instance, single-level evolutionary algorithms (EAs) were applied to subgroups containing only upper-level or lower-level variables. At the same time, a bilevel EA was employed for subgroups with both types of variables. Additionally, a criterion was introduced to address situations where multiple optima existed within specific subgroups, enhancing the algorithm's efficacy. The effectiveness of GO was validated through tests on scalable problems and its application to resource pricing in mobile edge computing. The article underscored the significance of considering interactions between variables in BOPs and provided a practical framework, offering a promising solution to address such challenges efficiently.

*Abdel-Basset et al. (2020)* introduced a novel approach termed HHOLS for energy-aware task scheduling in fog computing (TSFC), with a focus on enhancing QoS in Industrial Internet of Things (IIoT) applications. The methodology commenced by delineating a layered fog computing model, emphasizing its heterogeneous architecture to accommodate diverse computing resources effectively. To tackle the discrete TSFC problem, the standard Harris Hawks optimization algorithm was adapted through normalization and scaling techniques. Additionally, a swap mutation mechanism was employed to distribute workloads evenly among virtual machines, thereby enhancing solution quality. Notably, the integration of a local search strategy further augmented the performance of HHOLS. Comparative evaluations against other metaheuristic approaches across multiple performance metrics including energy consumption, makespan, cost, flow time, and carbon dioxide emission rate showcased the superior efficacy of HHOLS. The study underscored the pivotal role of fog computing in mitigating QoS challenges within IIoT applications, particularly amidst the escalating data influx to cloud computing. Furthermore, it emphasized the imperative of leveraging sustainable energy sources for fog computing servers to ensure long-term viability and efficiency.

*Yadav, Tripathi & Sharma (2022a)* introduced a hybrid approach for task scheduling in fog computing, merging a metaheuristic algorithm, fireworks algorithm (FWA), with a

heuristic algorithm known as heterogeneous earliest finish time (HEFT). Their combination aimed to minimize makespan and cost factors through bi-objective optimization. Experiments were conducted on distinct scientific workflows to compare the performance of the proposed BH-FWA algorithm against other approaches. Results from exhaustive simulations demonstrated the superiority of the BH-FWA algorithm in fog computing networks, as evidenced by metrics including makespan, cost, and throughput. The article contributed to the field by presenting a novel solution for task scheduling in fog computing environments and provided references for further exploration of related topics.

*Abdel-Basset et al. (2023)* introduced a pioneering multi-objective task scheduling strategy, denoted as M2MPA, founded on a modified marine predators algorithm (MMPA). This approach aimed to concurrently minimize energy consumption and make-span while adhering to the Pareto optimality theory. M2MPA enhanced the original MMPA by integrating a polynomial crossover operator to bolster exploration and an adaptive CF parameter to refine exploitation. The efficacy of M2MPA underwent assessment using eighteen tasks featuring varying scales and heterogeneous workloads allocated to two hundred fog devices. Through comparative experiments involving six established methodologies, M2MPA showcased significant superiority across diverse performance metrics, encompassing carbon dioxide emission rate, flowtime, make-span, and energy consumption. These findings underscored M2MPA's efficacy in optimizing task scheduling within fog computing environments, thus offering a promising avenue for adeptly managing data from IoT devices in cyber-physical-social systems.

*Abd Elaziz, Abualigah & Attiya (2021)* introduced AEOSSA, a modified artificial ecosystem-based optimization (AEO) technique integrated with Salp Swarm Algorithm (SSA) operators, aimed at improving task scheduling for IoT requests in cloud-fog environments. Through this modification, the algorithm's exploitation capability was enhanced, facilitating the search for optimal solutions. Evaluation of the AEOSSA approach utilized various synthetic and real-world datasets to assess its performance, comparing it against established metaheuristic methods. Results demonstrated the effectiveness of AEOSSA in tackling the task scheduling problem, surpassing other methods in metrics such as makespan time and throughput. This highlighted AEOSSA's potential as an efficient solution for scheduling IoT tasks in cloud-fog environments.

*Yadav, Tripathi & Sharma (2022b)* presented a modified fireworks algorithm that incorporated opposition-based learning and differential evolution approaches to address task scheduling optimization in fog computing environments. By utilizing the differential evolution operator, the algorithm aimed to overcome local optima, while opposition-based learning enhanced the diversity of the population's solution set. The method focused on minimizing both makespan and cost, thereby improving resource utilization efficiency. Through experiments conducted on various workloads, the performance of the proposed approach was compared with several recent metaheuristic approaches, demonstrating its efficacy in task scheduling optimization. The comparison underscored the significance of the proposed method in enhancing optimization outcomes.

*Abd Elaziz & Attiya (2021)* introduced a novel multi-objective approach, MHMPA, which integrated the marine predator's algorithm with the polynomial mutation

mechanism for task scheduling optimization in fog computing environments. This approach aimed to strike a balance between makespan and carbon emission ratio based on Pareto optimality, utilizing an external archive to store non-dominated solutions. Additionally, an improved version, MIMPA, using the Cauchy distribution and Levy Flight, was explored to enhance convergence and avoid local minima. Experimental results demonstrated the superiority of MIMPA over the standard version across various performance metrics. However, MHMPA consistently outperformed MIMPA even after integrating the polynomial mutation strategy. The efficacy of MHMPA was further validated through comparisons with well-known multi-objective optimization algorithms, showcasing significant improvements in flow time, carbon emission rate, energy, and makespan. Overall, the study highlighted the effectiveness of MHMPA in addressing task scheduling challenges in fog computing environments.

*Saif et al. (2023)* introduced a Multi-Objectives Grey Wolf Optimizer (MGWO) algorithm aimed at mitigating delay and reducing energy consumption in fog computing, particularly within fog brokers responsible for task distribution. Through extensive simulations, the efficacy of MGWO was verified, showcasing superior performance in mitigating delay and reducing energy consumption compared to existing algorithms. The article provided an overview of task scheduling challenges in cloud-fog computing and presented the MGWO algorithm as a solution. It compared MGWO with existing algorithms, highlighting its advantages in delay reduction and energy efficiency. The study delved into the mechanics of the MGWO algorithm, explaining its basis in the hunting behavior of grey wolves and detailing key components such as prey tracking, encircling, hunting, and attacking. Additionally, it discussed the algorithm's fitness function, mutation process, utilization of an external archive, and crowding distance. Simulation results affirmed the algorithm's efficacy in achieving its objectives.

*Abohamama, El-Ghamry & Hamouda (2022)* presented a semi-dynamic real-time task scheduling algorithm designed for bag-of-tasks applications within cloud-fog environments. This approach formulated task scheduling as a permutation-based optimization problem, leveraging a modified genetic algorithm to generate task permutations and allocate tasks to virtual machines based on minimum expected execution time. An optimality study demonstrated the algorithm's comparative performance with optimal solutions. Comparative evaluations against traditional scheduling algorithms, including first fit and best fit, as well as genetic and bees life algorithms, highlighted the proposed algorithm's superiority in terms of makespan, total execution time, failure rate, average delay time, and elapsed run time. The results showcased the algorithm's ability to achieve a balanced tradeoff between makespan and total execution cost while minimizing task failure rates, positioning it as an efficient solution for real-time task scheduling in cloud-fog environments.

*Nguyen et al. (2020)* presented a comprehensive approach to address task scheduling challenges in fog-cloud systems within the IoT context. It began by formulating a general model for the fog-cloud system, considering various constraints such as computation, storage, latency, power consumption, and costs. Subsequently, metaheuristic methods were applied to schedule data processing tasks, treating the problem as a multi-objective

optimization task. Simulated experiments validated the practical applicability of the proposed fog-cloud model and demonstrated the effectiveness of metaheuristic algorithms compared to traditional methods like Round-Robin. The study underscored the significance of fog computing in reducing latency and bandwidth usage while also emphasizing the crucial role of cloud services in providing high-performance computation and large-scale storage for IoT data processing.

*Mousavi et al. (2022)* discussed the importance of addressing the challenges posed by the rapid growth of data and latency-sensitive applications in the IoT era through efficient task scheduling in fog computing environments. The article introduced a bi-objective optimization problem focused on minimizing both server energy consumption and overall response time simultaneously. To address this challenge, the article proposed a novel approach called Directed Non-dominated Sorting Genetic Algorithm II (D-NSGA-II). This algorithm incorporates a recombination operator designed to strike a balance between exploration and exploitation capabilities. The algorithm's performance was evaluated against other meta-heuristic algorithms, demonstrating its superiority in meeting all requests before their deadlines while minimizing energy consumption. The study underscored the significance of fog computing in enhancing system performance and meeting the demands of IoT applications amidst the exponential growth of data.

*Salehnia et al. (2023)* proposed an IoT task request scheduling method using the Multi-Objective Moth-Flame Optimization (MOMFO) algorithm to enhance the quality of IoT services in fog-cloud computing environments. The approach aimed to reduce completion and system throughput times for task requests while minimizing energy consumption and $CO_2$ emissions. The proposed scheduling method was evaluated using datasets, and its performance was compared with other optimization algorithms, including PSO, FA, salp swarm algorithms (SSA), Harris Hawks optimizer (HHO), and artificial bee colony (ABC). Experimental results showed that the proposed solution effectively reduced task completion time, throughput time, energy consumption, and $CO_2$ emissions while improving the system's overall performance. The study highlighted the potential of MOMFO in optimizing task scheduling for IoT applications in fog-cloud environments.

*Nazeri, Soltanaghaei & Khorsand (2024)* proposed a predictive energy-aware scheduling framework for fog computing, integrating a MAPE-K control model comprising monitor, analyzer, planner, and executer components with a shared knowledge base. It introduced an Adaptive Network-based Fuzzy Inference System (ANFIS) in the Analyzer component to predict future resource load and a resource management strategy based on the predicted load to reduce energy consumption. Additionally, it combined the Improved Ant Lion Optimizer (ALO) and weighted GWO into a planner component called I-ALO-GWO for workflow scheduling. The framework's effectiveness was evaluated on IEEE CEC2019 benchmark functions and applied to scientific workflows using the iFogSim tool. Experimental results showed that I-ALO-GWO improved makespan, energy consumption, and total execution cost by significant percentages compared to alternative methods, addressing the inefficiencies and limitations of existing approaches in fog computing.

*Memari et al. (2022)* aimed to develop an infrastructure for smart home energy management at minimal hardware cost using cloud and fog computing, alongside proposing a latency-aware scheduling algorithm based on virtual machine matching employing meta-heuristics. Leveraging the effectiveness of Tabu search in various optimization problems, a novel algorithm enhanced with approximate nearest neighbor (ANN) and fruit fly optimization (FOA) algorithms were introduced. Through simulation and implementation of a case study, the algorithm's performance was evaluated considering execution time, latency, allocated memory, and cost function. Comparison outcomes revealed the superior performance of the proposed algorithm when compared to Tabu search, genetic algorithm, PSO, and simulated annealing methods. The research highlighted the importance of proficient task scheduling in both cloud and fog computing realms, emphasizing its role in optimizing resource allocation and diminishing response time in smart home energy management systems. This is particularly crucial in managing substantial data volumes while concurrently minimizing hardware expenses.

*Hussain & Begh (2022)* aimed to address the challenges of task scheduling in fog-cloud computing by proposing a novel algorithm called HFSGA (Hybrid Flamingo Search with a Genetic Algorithm) designed to minimize costs and enhance QoS. Utilizing seven benchmark optimization test functions, the performance of HFSGA was compared with other established algorithms. The comparison, validated through Friedman rank test, demonstrated the superiority of HFSGA in terms of percentage of deadline satisfied tasks (PDST), makespan, and cost. Comparative analysis against existing algorithms such as ACO, PSO, GA, Min-CCV, Min-V, and Round Robin (RR) showed HFSGA's effectiveness in optimizing task scheduling processes. The proposed algorithm was presented as a cost-efficient and QoS-aware solution tailored for fog-cloud environments, offering promising outcomes in improving task scheduling efficiency while minimizing associated costs. The article provided insights into fog-cloud system architecture and underscored the significance of HFSGA in addressing the complexities of task scheduling in such environments.

*Dev et al. (2022)* addressed the intricate task scheduling challenges arising from the increasing volumes of data in fog computing environments. In their study, they introduced a novel approach named Hybrid PSO and GWO (HPSO_GWO), which combines GWO with PSO. This hybrid meta-heuristic algorithm was developed to effectively allocate tasks to virtual machines (VMs) deployed across fog nodes. The research emphasized the importance of efficient task and resource scheduling in fog computing to ensure optimal QoS. Furthermore, the article referenced related work in the field, providing insights into the ongoing research endeavors aimed at addressing the challenges inherent in fog computing environments.

*Javaheri et al. (2022)* presented a multi-fog computing architecture tailored for optimizing workflow scheduling within IoT networks, with a focus on mitigating challenges related to fog provider availability and task scheduling efficiency. To address these concerns, the study introduced a hidden Markov model (HMM) designed to predict the availability of fog computing providers. This predictive model was trained using the unsupervised Baum-Welch algorithm and leveraged the Viterbi algorithm to compute fog

provider availability probabilities. Subsequently, these probabilities informed the selection of an optimal fog computing provider for scheduling IoT workflows. Additionally, the article proposed an enhanced version of the Harris hawks optimization (HHO) algorithm, termed discrete opposition-based HHO (DO-HHO), tailored specifically for scientific workflow scheduling. Extensive experiments conducted using iFogSim showcased significant reductions in tasks offloaded to cloud computing, instances of missed workflow deadlines, and SLA violations. The architecture comprised IoT networks, fog computing providers, and cloud computing data centers, with a broker node facilitating optimal fog computing resource selection. This approach aimed to enhance workflow scheduling efficiency in fog computing environments, outperforming existing state-of-the-art methods.

*Keshavarznejad, Rezvani & Adabi (2021)* addressed task offloading as a multi-objective optimization challenge targeting the reduction of total power consumption and task execution delay in the system. Acknowledging the NP-hard complexity of the problem, the researchers employed two meta-heuristic methods: the non-dominated sorting genetic algorithm (NSGA-II) and the Bees algorithm. Through simulations, both meta-heuristic approaches demonstrated resilience in minimizing energy consumption and task execution delay. The proposed techniques effectively balanced offloading probability with the power needed for data transmission, underscoring their efficacy in optimizing task offloading strategies within fog computing environments. The referenced articles covered a broad spectrum of research domains, including mobile cloud computing, fog computing, computation offloading, and optimization methodologies, offering a comprehensive overview of the field's current landscape and advancements.

*Hajam & Sofi (2023)* presented an innovative approach to fog computing, focusing on resource allocation and scheduling using the spider monkey optimization (SMO) algorithm. A heuristic initialization-based SMO algorithm was proposed to minimize the total cost of tasks by selecting optimal fog nodes for computation offloading. Three initialization strategies-longest job fastest processor (LJFP), shortest job fastest processor (SJFP), and minimum completion time (MCT)-were compared in terms of average cost, service time, monetary cost, and cost per schedule. Results indicated the superiority of the MCT-SMO approach over other heuristic-based SMO algorithms and PSO in terms of efficiency and effectiveness in fog computing environments. The study provided valuable insights into enhancing task scheduling and resource allocation in fog computing networks.

*Hussein & Mousa (2020)* introduced two nature-inspired meta-heuristic schedulers, ACO and PSO, aimed at optimizing task scheduling and load balancing in fog computing environments for IoT applications, particularly in intelligent city scenarios. The proposed algorithms considered communication cost and response time to distribute IoT tasks across fog nodes effectively. Experimental evaluations compared the performance of the ACO-based scheduler with the PSO-based and RR algorithms. Results demonstrated that the ACO-based scheduler improved response times for IoT applications and achieved efficient load balancing among fog nodes compared to the PSO-based and RR algorithms. The study highlighted the potential of nature-inspired meta-heuristic algorithms in

enhancing task scheduling in fog computing environments and outlined future research directions in multi-objective optimization and dynamic IoT scenarios.

*Yadav, Tripathi & Sharma (2023)* explored the importance of efficient task scheduling in fog computing networks, particularly for minimizing service time and enhancing stability in dynamic fog devices. It introduced the opposition-based chemical reaction (OBCR) method, a novel approach that combined heuristic ranking, chemical reaction optimization (CRO), and opposition-based learning (OBL) approaches. OBCR aimed to optimize task scheduling by ensuring better exploration and exploitation of the solution space while escaping local optima. The OBCR method utilized four operators to enhance the stability of dynamic fog devices and optimize task scheduling in fog computing networks. Through extensive simulations, OBCR exhibited superior performance over alternative approaches, effectively reducing service-time latency and improving network stability. This contribution addresses significant challenges in fog computing task scheduling, offering a robust solution with the OBCR method.

*Abdel-Basset et al. (2020)* introduced a novel approach to task scheduling in fog computing, aiming to enhance the QoS for IoT applications by offloading tasks from the cloud. It proposed an energy-aware model called the marine predators algorithm (MPA) for fog computing task scheduling (TSFC). Three versions of MPA were presented, with the modified MPA (MMPA) demonstrating superior performance over other algorithms. The study evaluated various performance metrics, including energy consumption, makespan, flow time, and carbon dioxide emission rate. It emphasized the significance of fog computing in improving QoS for IoT applications, particularly in domains such as healthcare and smart cities, by optimizing task scheduling strategies.

*Kishor & Chakarbarty (2022)* proposed a meta-heuristic task offloading algorithm, Smart Ant Colony Optimization (SACO), inspired by nature, for offloading IoT-sensor applications tasks in a fog environment. The proposed algorithm was compared with RR, the throttled scheduler algorithm, two bio-inspired algorithms, modified particle swarm optimization (MPSO), and bee life algorithm (BLA). Numerical results demonstrated a significant improvement in latency with the SACO algorithm compared to RR, throttled, MPSO, and BLA, reducing task offloading time by 12.88%, 6.98%, 5.91%, and 3.53%, respectively. The study highlighted the effectiveness of SACO in optimizing task offloading for IoT-sensor applications in fog computing environments, offering potential enhancements in latency reduction and overall system performance.

*Khaledian et al. (2024)* proposed a hybrid particle swarm optimization and simulated annealing algorithm (PSO-SA) for task prioritization and optimizing energy consumption and makespan in fog-cloud environments. The proposed approach targeted the task scheduling complexities prevalent in the IoT environment, aiming to enhance operational efficiency and overall performance. Simulation results showcased notable enhancements achieved by the PSO-SA algorithm, with a 5% reduction in energy consumption and a 9% improvement in makespan compared to the baseline algorithm (IKH-EFT). These findings underscored the pivotal role of hybrid optimization strategies in optimizing task allocation and system performance within fog and cloud computing environments.

*Khiat, Haddadi & Bahnes (2024)* addressed task scheduling in a fog-cloud environment and proposed a novel genetic-based algorithm called GAMMR to optimize the balance between total energy consumption and response time. Through simulations conducted on eight datasets of varying sizes, the proposed algorithm's performance was evaluated. Results indicated that the GAMMR algorithm consistently outperformed the standard genetic algorithm, achieving an average improvement of 3.4% in the normalized function across all tested cases. The references provided in the text offered a comprehensive overview of fog-cloud services, IoT applications, task scheduling strategies, and optimization algorithms in cloud and fog computing environments, contributing to a deeper understanding of current research trends and challenges in the field.

*Ahmadabadi, Mood & Souri (2023)* evaluated the performance of a newly proposed algorithm in two scenarios. Firstly, the algorithm was compared with several popular multi-objective optimization methods using standard test functions, showing superior performance. Secondly, the algorithm was applied to solve the task scheduling problem in fog-cloud systems within the IoT. The approach incorporated a multi-objective function aiming to minimize makespan, energy consumption, and monetary cost. Additionally, a new operator called star-quake was introduced in the multi-objective version of the gravitational search algorithm (MOGSA) to enhance performance. The results demonstrated significant improvements, including an 18% reduction in makespan, 22% decrease in energy consumption, and 40% reduction in processing cost. Statistical analysis further confirmed the algorithm's superiority over other approaches in task scheduling. The article contributed to addressing the challenges of task scheduling in fog-cloud systems and highlighted the effectiveness of the proposed algorithm in optimizing system performance metrics.

Table 5 provides a comparative summary of studies focusing on metaheuristic approaches for task scheduling.

## Learning-based heuristics

Learning-based heuristics represented a paradigm shift in task scheduling within fog computing, harnessing the power of machine learning to create dynamic and adaptable algorithms (*Memari et al., 2022*; *Fahimullah et al., 2023*; *Ibrahim & Askar, 2023*). Unlike traditional methods, learning-based approaches observed fog environments, collected data on resource availability and task characteristics, and dynamically adjusted scheduling strategies based on learned insights. Utilizing approaches like reinforcement learning (RL), Q-learning, and Deep Q-Networks (DQN), these heuristics adapt to changing conditions, optimize resource allocation, and improve overall efficiency. Despite challenges such as the need for extensive training data and computational overhead, learning-based heuristics offered significant potential for revolutionizing fog computing, enabling more efficient and intelligent task scheduling to meet the evolving demands of IoT applications in fog environments.

*Fellir et al. (2020)* proposed a multi-agent-based model for task scheduling in cloud-fog computing environments to address the challenges of managing large volumes of data generated by IoT devices. The model prioritized tasks based on factors such as task

**Table 5 Comparison of metaheuristic studies for task scheduling.**

| Study | General approach | Performance & optimization | Implementation & evaluation | Performance metrics | Strengths | Limitations |
|---|---|---|---|---|---|---|
| Hosseinioun et al. (2020) | Energy-aware method using DVFS technique and hybrid IWO-CA algorithm | Reduction in energy consumption; Constructing valid task sequences | Experimental validation; Hybrid algorithm effectiveness | Energy consumption reduction | Enhanced scheduling | Limited scalability analysis |
| Hosseini, Nickray & Ghanbari (2022) | PQFAHP algorithm combining Priority Queue, Fuzzy logic, and AHP | Multi-criteria task prioritization; Dynamic scheduling | Experimental comparison; Algorithm superiority | Waiting time, delay, service level, mean response time, number of scheduled tasks | Improved multi-criteria scheduling | Lack of scalability analysis |
| Apat et al. (2019) | Mapping independent tasks to fog layer; Priority based on deadlines | Better performance compared to cloud; Three-layer architecture | Comparative evaluation; Performance metrics | CPU, RAM utilization; Task completion time | Better fog layer performance | Lack of extensive scalability analysis |
| Jalilvand Aghdam Bonab & Shaghaghi Kandovan (2022) | QoS-aware resource allocation in MEC; Optimization of energy efficiency | MINLP optimization; Carrier allocation strategy | Simulation-based performance evaluation | Energy efficiency, network throughput | Improved energy efficiency; Multi-source scenarios | Limited discussion on scalability |
| Huang & Wang (2020) | GO framework for large-scale BOPs; Phase-wise optimization | Bilevel optimization; Resource pricing | Validation on scalable problems; Application to edge computing | Reduction in optimization complexity | Practical framework | Limited discussion on real-world applicability |
| Abdel-Basset et al. (2020) | HHOLS algorithm based on Harris Hawks optimization | Energy-aware task scheduling; Local search strategy | Comparative analysis; Superior performance | Energy consumption, makespan, cost, flow time | Superior performance over other metaheuristics | Lack of real-world deployment analysis |
| Yadav, Tripathi & Sharma (2022a) | Hybrid BH-FWA algorithm merging FWA with HEFT | Bi-objective optimization; Task scheduling | Comparative experiments; Performance metrics | Makespan, cost, throughput | Improved makespan, cost factors | Limited scalability analysis |
| Abdel-Basset et al. (2023) | M2MPA algorithm based on modified MMPA | Multi-objective optimization; Pareto optimality | Comparative experiments; Performance indicators | Carbon emission rate, flowtime, make-span, energy consumption | Substantial superiority over other approaches | Limited scalability analysis |
| Abd Elaziz, Abualigah & Attiya (2021) | AEOSSA algorithm integrating AEO with SSA | Task scheduling optimization; Exploitation capability | Performance evaluation; Comparison with metaheuristics | Makespan time, throughput | Efficient scheduling; Outperforms other methods | Lack of real-world deployment analysis |
| Yadav, Tripathi & Sharma (2022b) | Enhanced fireworks algorithm with OBL and DE | Task scheduling optimization; Minimization of makespan and cost | Comparative experiments; Efficacy evaluation | Makespan, cost | Improved resource utilization efficiency | Limited discussion on real-world applicability |
| Abd Elaziz & Attiya (2021) | MHMPA algorithm integrating MMPA with mutation strategies | Multi-objective optimization; Pareto optimality | Comparative experiments; Performance metrics | Flow time, carbon emission rate, energy, makespan | Effective optimization; Superior performance | Limited scalability analysis |

| Study | General approach | Performance & optimization | Implementation & evaluation | Performance metrics | Strengths | Limitations |
|---|---|---|---|---|---|---|
| *Saif et al. (2023)* | MGWO algorithm for delay and energy reduction | Task distribution optimization; Energy efficiency | Comparative analysis; Performance metrics | Delay, energy consumption | Superior performance over existing algorithms | Limited scalability analysis |
| *Abohamama, El-Ghamry & Hamouda (2022)* | Real-time task scheduling algorithm for bag-of-tasks | Permutation-based optimization; Genetic algorithm | Comparative evaluations; Performance metrics | Makespan, execution time, failure rate, delay time | Balanced trade-off; Efficient scheduling | Limited scalability analysis |
| *Nguyen et al. (2020)* | Fog-cloud model with metaheuristic scheduling | Multi-objective optimization; Fog-cloud system model | Simulation-based validation; Comparative evaluation | Computation, storage, latency, power consumption | Effective fog-cloud modeling; Improved optimization | Limited real-world deployment analysis |
| *Mousavi et al. (2022)* | D-NSGA-II algorithm for energy consumption and response time | Bi-objective optimization; Minimization of energy consumption | Experimental validation; Comparison with metaheuristics | Energy consumption, response time | Effective optimization; Enhanced system performance | Limited scalability analysis |
| *Salehnia et al. (2023)* | MOMFO algorithm for IoT task scheduling | Multi-objective optimization; IoT service enhancement | Comparative evaluation; Performance metrics | Completion time, throughput, energy, $CO_2$ emissions | Improved system performance; Efficient scheduling | Lack of real-world deployment analysis |
| *Nazeri, Soltanaghaei & Khorsand (2024)* | Predictive energy-aware scheduling framework | MAPE-K control model with ANFIS; Workflow scheduling | Simulation-based validation; Performance metrics | Makespan, energy consumption, total cost | Improved optimization outcomes; Enhanced scheduling | Limited scalability analysis |
| *Memari et al. (2022)* | Latency-aware scheduling algorithm with Tabu search | Tabu search with ANN and FOA; Smart home energy management | Case study implementation; Comparative evaluation | Execution time, latency, memory, cost | Superior performance over other methods | Lack of real-world deployment analysis |
| *Hussain & Begh (2022)* | HFSGA algorithm for cost-efficient task scheduling | Hybrid Flamingo Search with GA; Fog-cloud environments | Comparative analysis; Performance metrics | Percentage of Deadline Satisfied Tasks, makespan, cost | Cost-efficient scheduling; QoS-aware solution | Limited scalability analysis |
| *Dev et al. (2022)* | HPSO_GWO algorithm for task scheduling | Hybrid PSO and GWO; Fog computing environments | Contribution to task and resource scheduling | N/A | Optimization of QoS; Insights into fog computing challenges | Lack of real-world deployment analysis |
| *Javaheri et al. (2022)* | HMM-based fog provider availability prediction with DO-HHO algorithm | Hidden Markov Model with DO-HHO; IoT workflow scheduling | Simulation-based evaluation; Performance metrics | Offloaded tasks, missed deadlines, SLA violations | Streamlined workflow scheduling; Superior performance | Lack of real-world deployment analysis |
| *Keshavarznejad, Rezvani & Adabi (2021)* | Multi-objective optimization with NSGA-II and Bees algorithm | Task offloading optimization; Fog computing environments | Simulation-based evaluation; Performance metrics | Energy consumption, delay | Efficient task offloading; Favorable tradeoff | Limited scalability analysis |

| Study | General approach | Performance & optimization | Implementation & evaluation | Performance metrics | Strengths | Limitations |
|---|---|---|---|---|---|---|
| *Hajam & Sofi (2023)* | SMO algorithm for resource allocation and scheduling | Spider monkey optimization; Fog computing environments | Comparative evaluation; Performance metrics | Average cost, service time, monetary cost | Efficient resource allocation; Superior performance | Lack of real-world deployment analysis |
| *Hussein & Mousa (2020)* | ACO and PSO-based schedulers for IoT task distribution | Ant colony and particle swarm optimization; Fog computing environments | Experimental validation; Comparison with RR algorithm | Response times, load balancing | Effective load balancing; Improved response times | Limited scalability analysis |
| *Yadav, Tripathi & Sharma (2023)* | OBCR method for task scheduling in fog networks | Opposition-based chemical reaction; Fog computing environments | Simulation-based evaluation; Performance metrics | Service-time latency, stability | Robust solution; Improved stability | Lack of real-world deployment analysis |
| *Abdel-Basset et al. (2020)* | MPA-based algorithm for fog task scheduling | Marine predators algorithm; Fog computing environments | Comparative analysis; Performance metrics | Energy consumption, makespan, flow time, carbon dioxide emission rate | Improved QoS; Effective scheduling | Limited scalability analysis |
| *Kishor & Chakarbarty (2022)* | SACO algorithm for IoT-sensor task offloading | Smart ant colony optimization; Fog computing environments | Performance comparison; Latency reduction | Task offloading time | Latency reduction; System performance enhancement | Lack of real-world deployment analysis |
| *Khaledian et al. (2024)* | PSO-SA algorithm for energy optimization in fog-cloud | Hybrid PSO and simulated annealing; Fog-cloud environments | Comparative evaluation; Performance metrics | Energy consumption, makespan | Improvement in energy consumption, makespan | Limited scalability analysis |
| *Khiat, Haddadi & Bahnes (2024)* | GAMMR algorithm for energy consumption and response time | Genetic-based algorithm; Fog-cloud environments | Experimental validation; Performance metrics | Energy consumption, response time | Improved optimization; Balanced performance | Lack of real-world deployment analysis |
| *Ahmadabadi, Mood & Souri (2023)* | MOGSA algorithm with star-quake operator for fog-cloud task scheduling | Multi-objective gravitational search algorithm; Fog-cloud environments | Evaluation in two scenarios; Performance metrics | Makespan, energy consumption, processing cost | Significant performance improvements | Lack of real-world deployment analysis |

priority, wait time, status, and resource requirements, aiming to serve the most critical tasks first. It also updated the priority value of tasks, considering their dependencies on other tasks and their priorities. Simulation results demonstrated that the proposed model improved resource utilization and overall system performance. The article highlighted the advantages of using multi-agent systems in cloud-fog platforms and suggested the underutilization of this approach in current systems.

*Wang et al. (2024)* introduced a novel Deep Reinforcement Learning-based IoT application Scheduling algorithm, DRLIS, designed to optimize the response time of heterogeneous IoT applications and balance the load on edge and fog servers efficiently. Implemented within the FogBus2 function-as-a-service framework, DRLIS aimed to create

an integrated edge-fog-cloud serverless computing environment. Through extensive experiments, the study demonstrated that DRLIS significantly reduced the execution cost of IoT applications, achieving up to 55%, 37%, and 50% improvements in load balancing, response time, and weighted cost, respectively, compared to other reinforcement learning approaches and metaheuristic algorithms. The research underscored the effectiveness of DRLIS in improving system performance and efficiency in edge and fog computing environments, showcasing its potential for real-world deployment.

In a study (*Liu et al., 2016*), they employed a Markov decision process (MDP) strategy to mitigate average task execution latency in edge computing environments. Their approach introduced a streamlined one-dimensional search algorithm to determine the optimal task scheduling policy. However, limitations arose in adapting to dynamic computing environments and addressing complex weighted cost optimization challenges typical of heterogeneous fog computing scenarios. In a separate investigation, *Wu et al. (2018)* conceptualized task scheduling problems in edge and fog computing contexts as directed acyclic graphs (DAGs) (*Wu et al., 2018*). They proposed an estimation of distribution algorithm (EDA) coupled with a partitioning operator to facilitate task queuing and server assignment within the graph structure. Nevertheless, practical implementation and testing of their methodology were not conducted.

*Sun, Lin & Xu (2018)* improved the NSGA2 algorithm and developed a resource scheduling strategy for fog nodes within a single fog cluster, taking into account the diverse characteristics of different devices. While their efforts aimed at reducing service latency and ensuring task execution stability, they were confined to addressing scheduling challenges within a singular computing environment. Similarly, *Ali et al. (2020)* advocated for an NSGA2-based approach to minimize total computation time and system cost in task scheduling across heterogeneous fog-cloud environments. Their optimization framework aimed to dynamically allocate resources for predefined tasks. Nonetheless, akin to previous studies, their reliance on metaheuristic algorithms assumed a certain level of task awareness for the development of scheduling policies, thereby limiting adaptability to dynamic and intricate scenarios.

*Ramezani Shahidani et al. (2023)* introduced a Q-learning-based algorithm aimed at improving task execution latency and load balancing within fog-cloud computing environments. However, this method did not account for inter-task dependencies and the diverse characteristics inherent in fog and cloud computing setups. *Baek et al. (2019)* further developed the Q-learning algorithm to handle load balancing specifically in fog computing, considering node diversity. Nevertheless, it made assumptions about task independence within applications. *Jie et al. (2021)* proposed a Deep Q-Network (DQN) approach to minimize task processing latency in edge computing, treating task scheduling as a Markov decision process while recognizing the diversity of IoT applications. However, its scope was limited to edge computing and a single optimization objective. *Xiong et al. (2020)* also utilized DQN for resource allocation in IoT edge systems, with a focus on minimizing job completion time, but without addressing multi-objective optimization.

*Wang et al. (2019a)* proposed a DRLRA scheme for edge computing, targeting reduced service time and resource utilization balance but without considering fog computing or

practical implementation. *Huang et al. (2019)* utilized DQN for edge resource allocation, emphasizing weighted cost reduction without addressing server heterogeneity or task dependencies. *Chen et al. (2018)* proposed a double DQN approach for edge task execution and energy balance, limited to edge environments and ignoring task dependencies. *Zheng et al. (2022b)* introduced an algorithm based on soft actor-critic (SAC) for minimizing edge task completion time, utilizing simulation-based experiments. *Zhao, Li & He (2023)* proposed a TD3-based deep reinforcement learning (DRL) algorithm to reduce latency and energy consumption, although they overlooked task dependencies and relied solely on simulations. *Liao et al. (2023)* applied deep deterministic policy gradient (DDPG) and double deep Q-network (DQN) algorithms in edge computation to minimize energy consumption and latency. However, their approach did not consider fog environments or device heterogeneity. *Sethi & Pal (2023)* introduced a DQN algorithm for optimizing fog server energy consumption and load balancing. However, their method was validated only through simulations and did not account for task dependencies.

*Wang et al. (2019b)* explored the potential of fog computing (FC) in supporting computation-intensive applications for future wireless networks and underscored the efficacy of nonorthogonal multiple access (NOMA) for enhanced spectrum efficiency. They presented a NOMA-based FC system where multiple task nodes employed NOMA for task scheduling to a helper node with abundant computation resources. The overarching objective was to minimize the sum cost, encompassing delay and energy consumption for all task nodes, thereby achieving an energy-delay tradeoff. Given the complexity of this combinatorial optimization problem, the article introduced an online learning-based optimization framework to address it. Simulation outcomes highlighted the significant reduction in the sum cost achieved by the proposed scheme compared to baseline approaches. The study underscored the importance of leveraging the capacity of the multiple access channel in task scheduling and execution to optimize energy-delay tradeoffs in fog computing systems.

*Ibrahim & Askar (2023)* introduced an intelligent scheduling strategy for fog computing (FC) systems, leveraging multi-objective optimization algorithms, including multi-objective evolutionary algorithm based on decomposition (MOEA/D) and non-dominated sorting genetic algorithm (NSGA2), to select optimal nodes based on three objectives: node load, distance, and task priority. This approach termed the multi-objective deep reinforcement learning (MODRL) algorithm, integrated a Deep Q Network (DQN) with multi-objective optimization approaches to enhance task allocation and scheduling in FC environments. Simulation results showcased the superiority of the proposed strategy across various performance metrics, including task completion time, makespan, queueing delay, and CPU load, among others, with significant average improvements compared to existing research studies. The proposed system architecture comprised three layers: IoT layer, FC layer, and cloud computing layer, with detailed environment dynamics, algorithm design, and evaluation metrics provided. The results underscored the effectiveness and adaptability of the MODRL-based approach in optimizing task processing in FC environments. The article concluded by highlighting future research

directions, particularly in enhancing task re-scheduling mechanisms to address long queuing times in the orchestrator.

*Gao et al. (2020)* introduced a collaborative computing framework integrating local computing (mobile device), edge cloud (MEC), and central cloud (MCC) components to enhance resource allocation and computation offloading for tasks with high computational requirements. Within this framework, they devised a novel Q-learning-based computation offloading (QLCOF) policy aimed at achieving optimal resource allocation and offloading decisions through pre-scheduled task computations from a global perspective. The decision-making process for offloading was modeled as a Markov decision process (MDP), with a state loss function (STLF) designed to assess the quality of experience (QoE). Subsequently, a system loss function (SYLF) was formulated, and a scheme was developed to minimize it by optimizing transmission power and computation frequency. An offloading scheme based on Q-learning (QLOF) was then implemented using pre-calculated transmission power and computation frequency, resulting in a reduction of the SYLF across various parameters according to numerical simulations. This proposed system and algorithm offer a comprehensive approach to optimizing computation offloading decisions within collaborative computing environments, with potential applications in enhancing overall system performance and efficiency.

*Siyadatzadeh et al. (2023)* presented a novel primary backup task assignment strategy, ReLIEF, to enhance the reliability of fog-based IoT systems. By utilizing machine learning approaches, particularly reinforcement learning (RL), ReLIEF identified suitable nodes for executing primary and backup tasks, effectively balancing communication delay and workload on fog devices. Simulation results demonstrated that ReLIEF significantly reduced task dropping rates by up to 84% compared to existing approaches. Additionally, it improved workload distribution and system reliability by nearly 72%, addressing the challenges of real-time task execution in fog computing environments. With the proliferation of IoT devices and the associated data volume, ReLIEF offered a promising solution to enhance system performance and reliability in fog-based IoT systems.

*Gazori, Rahbari & Nickray (2020)* focused on addressing the challenge of efficient task scheduling in fog-based IoT applications to minimize long-term service delay and computation cost while adhering to resource and deadline constraints. To address this challenge, the study utilized a reinforcement learning methodology and introduced a novel scheduling algorithm based on DDQL, incorporating target network and experience replay techniques. Assessment outcomes revealed that the proposed algorithm outperformed conventional methods across several metrics, including service delay, computational expenses, energy usage, and task completion rates. Moreover, the algorithm adeptly managed issues such as single point of failure (SPoF) and load distribution within the fog environment. The article provided a comprehensive overview of related work, outlined the system model and problem formulation, presented the proposed algorithm, and discussed the simulation environment. Overall, the study contributed valuable insights into optimizing task scheduling in fog-based IoT applications, although further research may be needed to address potential limitations and explore additional enhancements.

*Guevara et al. (2022)* proposed three multi-objective task scheduling algorithms tailored for the cloud-fog continuum, aiming to minimize both workflow makespan and processing cost while accommodating the QoS demands of applications. It acknowledged the complexity of task scheduling in this environment due to diverse application requirements and device capabilities, with conflicting objectives of minimizing makespan and processing cost exacerbating the challenge. The proposed algorithms leveraged reinforcement learning approaches to address these complexities and optimize task scheduling decisions. Numerical simulations demonstrated the superiority of the reinforcement learning-based scheduler over classical optimization-based approaches, showcasing its effectiveness in achieving better scheduling outcomes. However, the article acknowledged potential limitations inherent in the proposed algorithms and suggested avenues for future research to address them and further enhance scheduling efficiency in cloud-fog environments.

*Tahmasebi-Pouya, Sarram & Mostafavi (2023)* proposed a novel approach for fair load distribution in fog computing environments, utilizing a Q-learning algorithm-based load balancing method executed at the fog layer. The objective was to enhance load balancing and reduce delay simultaneously. In this method, fog nodes employ reinforcement learning to decide whether to process tasks locally from IoT devices or offload them to nearby fog nodes or the cloud. Simulation results demonstrated that the proposed approach achieved superior load distribution among fog nodes, improving delay, runtime, network utilization, and load standard deviation compared to other approaches. For instance, with four fog nodes, the proposed method reduced delay by approximately 8.44% compared to the load balancing and optimization strategy (LBOS), 26.65% compared to secure authentication and load balancing (SALB), 29.15% compared to the proportional method, 7.75% compared to fog cluster-based load balancing (FCBLB), and 36.22% compared to the random method. Similarly, with ten fog nodes, the proposed method reduced delay by about 13.80% compared to LBOS, 29.84% compared to SALB, 32.23% compared to the proportional method, 13.34% compared to FCBLB, and 39.1% compared to the random method. The article contributed valuable insights into load-balancing algorithms and their efficacy in fog computing environments, offering potential avenues for further research and optimization.

*Raju & Mothku (2023)* introduced a task scheduling strategy for fog-enabled IoT architecture to address the challenges posed by limited computing resources and increasing demand for executing complex and deadline-aware tasks. The proposed strategy utilized a fuzzy-based reinforcement learning (FRL) mechanism to prioritize tasks based on fuzzy logic and schedule them using an on-policy reinforcement learning technique. This approach aimed to reduce service delay and energy consumption of fog nodes. By formulating the scheduling problem as mixed-integer nonlinear programming (MINLP), the proposed technique achieved improved long-term rewards compared to traditional–learning approaches. Evaluation results indicated that the proposed task scheduling technique outperformed existing algorithms, demonstrating improvements of up to 23% and 18% in service latency and energy consumption, respectively. The article underscored the significance of efficient task scheduling in fog computing environments and provided valuable insights into mitigating latency and energy consumption challenges.

*Farhat, Sami & Mourad (2020)* introduced a fog scheduling decision model based on reinforcement learning (R-learning) to optimize fog placement and decrease the cloud's load in fog computing environments. The model focused on studying service requester behavior and generating a suitable fog placement schedule based on average reward. An implementation of the R-learning model was provided, followed by experiments on a real dataset to demonstrate its efficiency in utilizing fog resources and reducing the cloud's load. The model's ability to adapt to changing user demands over time was also highlighted. Comparative experiments with two other fog placement approaches (threshold-based and random-based) showed a significant decrease in the number of processed requests by the cloud, ranging from 100% to 30%, with a limited number of fogs to deploy. These findings underscored the importance of the proposed fog scheduling decision model in effectively placing on-demand fog resources to meet user needs while optimizing resource utilization and reducing cloud dependency.

*Shi et al. (2020)* introduced a novel approach to task offloading within vehicular fog computing (VFC), which aims to encourage vehicles to contribute their idle computing resources. Their method incorporates dynamic pricing mechanisms that take into account factors such as vehicle mobility, task priority, and service availability to incentivize resource sharing. The task offloading challenge was framed as a Markov decision process (MDP), focusing on maximizing the mean latency-aware utility of tasks over time. To tackle this problem, the researchers developed a soft actor-critic (SAC)-based deep reinforcement learning (DRL) algorithm. This algorithm was designed to simultaneously maximize the expected reward and policy entropy. Through extensive simulations, the effectiveness and superiority of their proposed scheme were confirmed when compared to conventional algorithms, highlighting its potential to enhance computational capabilities within VFC environments.

*Jain & Kumar (2023)* investigated the task offloading challenge in fog computing environments, emphasizing user QoS metrics like end-to-end latency, energy consumption, task deadlines, and priority. They formulated the problem as a MDP to optimize resource allocation. Three model-free off-policy DRL solutions were proposed to maximize rewards, focusing on resource utilization. Extensive experimentation was conducted to validate and compare the efficacy of these proposed mechanisms. Results revealed that the proposed approach achieved an average satisfaction rate of 96.23% for task deadlines, indicating an 8.25% enhancement in efficiency. The references cited encompassed a range of topics in fog computing, edge computing, and IoT, offering valuable insights and solutions to address challenges in these domains.

*Mishra et al. (2023)* investigated task offloading approaches in VFC networks and proposed a federated learning-supported Deep Q-Learning-based (FedDQL) approach for optimal task offloading within a collaborative computing framework. The method considered factors such as latency and energy consumption in computations, communications, offloading, and resource utilization. It addressed the tradeoffs between latency and computing/communication constraints. The efficacy of the FedDQL scheme was evaluated for dependent task sets, demonstrating its effectiveness through extensive simulations. Results showed that the proposed method outperformed others, achieving an

average improvement of 49%, 34.3%, 29.2%, 16.2%, and 8.21% across various performance metrics. Overall, the study provided insights into optimizing task offloading in VFC networks, highlighting the strengths of the proposed FedDQL approach in improving service rates, utilization, energy consumption, and latency rates compared to alternative strategies.

*Yeganeh, Sangar & Azizi (2023)* introduced a novel hybrid optimization algorithm, E-AEO-AOA, designed to address resource limitations and task deadlines in mobile edge computing (MEC) networks comprising smart mobile devices (SMDs), fog nodes, and cloud resources. The approach combined Artificial Ecosystem-based Optimization (AEO) and Arithmetic Optimization Algorithm (AOA) while integrating enhancements such as Q-learning and chaos theory. The algorithm aimed to minimize execution time and energy consumption by optimizing task offloading and scheduling. Experimental evaluation conducted on fifteen MEC networks compared E-AEO-AOA with various existing algorithms statistically, revealing its superiority in 90% of cases. Additionally, visual comparisons of convergence rates and solution dispersity, as well as Wilcoxon signed-rank tests, further supported the algorithm's effectiveness. The article provided comprehensive insights into related works, the system model, problem formulation, algorithmic details, and experimental outcomes, concluding that E-AEO-AOA was a promising solution for optimizing task offloading and scheduling in MEC networks.

*Fahimullah et al. (2023)* conducted a thorough analysis of existing literature concerning machine learning-based approaches to tackle resource management challenges within fog computing. These challenges encompassed diverse aspects such as resource provisioning, application placement, scheduling, allocation, task offloading, and load balancing. Through a meticulous examination, the literature was compared based on the strategies, metrics, tools, datasets, and approaches employed. Furthermore, the study identified pertinent research gaps in resource management and proposed future directions to advance the field. The article contributed a comprehensive overview of the topic, supported by a wealth of references and affiliations of the authors, thereby serving as a valuable resource for researchers and practitioners in the domain.

*Devarajan et al. (2023)* presented a novel two-stage deep reinforcement learning (DRL) based system tailored for smart agriculture. In the initial stage, an ACO enabled DQN model, termed MACO-DQN, was introduced to efficiently offload tasks such as fire detection, pest detection, and crop growth monitoring to edge, fog, or cloud devices based on latency, energy consumption, and computing power. Following task offloading, the second stage utilized a DRL-based DQN model, referred to as RL-DQN, for monitoring and predicting various agricultural activities. Experimental findings showcased notable enhancements in convergence speed, planning success rate, and path accuracy compared to traditional deep Q-networks-based intensive learning methods. Performance evaluation metrics, including precision, recall, F-measure, and accuracy, demonstrated the superiority of the proposed methodology, with results indicating a precision of 98.5%, recall of 99.1%, F-measure of 98.1%, and accuracy of 98.5%. The study also underscored the integration of IoT, unmanned aerial vehicles (UAV), and edge-fog-cloud architecture in enabling smart agriculture solutions to meet the challenges of global food demand.

*Zheng et al. (2022a)* proposed a DRL-based approach to workload scheduling in edge computing, aiming to address challenges associated with dynamic environments. By adopting DQN algorithms, the approach tackled the complexity and high dimensionality of the workload scheduling problem. The goal was to balance workload, reduce service time, and minimize the failed task rate. Through extensive simulations, the proposed approach demonstrated superior performance compared to other methods, particularly in terms of service time, virtual machine (VM) utilization, and failed task rate. The DRL-based solution offered efficiency and effectiveness in addressing workload scheduling challenges in edge computing, providing a promising avenue for further research and implementation.

*Sellami et al. (2022)* investigated the challenge of energy consumption and latency in Software-Defined Fog-IoT Networks, proposing an energy-aware and low-latency task scheduling solution. Initially, the problem was formulated as an energy-constrained Deep Q-Learning process to minimize network latency while ensuring energy efficiency under application constraints. Then, a DRL approach was introduced for dynamic task scheduling and assignment in SDN-enabled edge networks. Through comprehensive experiments and comparisons with three pioneering deep learning algorithms, the proposed solution demonstrated superior performance, achieving better energy-saving by up to 87% and enabling more task assignments with up to 50% less time delay. The study provided a thorough analysis of related work, a detailed model for the task assignment and scheduling problem, and a performance evaluation. It also acknowledged funding support and declared affiliations and research interests.

*Jayanetti, Halgamuge & Buyya (2022)* addressed the challenge of workflow scheduling in edge-cloud environments for IoT applications, aiming to balance energy consumption and execution time. It introduced a novel hierarchical action space and a hybrid DRL model for efficient task scheduling. The proposed framework was evaluated against baseline algorithms using metrics such as energy consumption, execution time, percentage of deadline hits, and percentage of completed jobs. Results indicated that the DRL technique performed significantly better, achieving a 56% improvement in energy consumption and a 46% improvement in execution time compared to optimized baselines. Moreover, it maintained energy efficiency and execution time on par with respective optimized baselines, demonstrating its superior ability to handle the conflicting goals of minimizing energy consumption and execution time in workflow scheduling across edge-cloud environments.

Table 6 provides a comparative summary of studies focusing on learning-based heuristic approaches for task scheduling.

## Hybrid heuristics

Hybrid heuristics presented a promising avenue for addressing the intricate challenges of task scheduling in fog computing environments (*Dubey & Sharma, 2023*; *Kaushik & Al-Raweshidy, 2022*). By amalgamating the strengths of individual heuristic approaches, these hybrid approaches offered heightened adaptability, solution quality, and efficiency. Strategies such as combining greedy methods for swift initial solutions with subsequent

Alsadie (2024), *PeerJ Comput. Sci.*, DOI 10.7717/peerj-cs.2128

**Table 6 Comparison of learning-based heuristic studies for task scheduling.**

| Study | General approach | Performance & optimization | Implementation & evaluation | Performance metrics | Strengths | Limitations |
|---|---|---|---|---|---|---|
| *Fellir et al. (2020)* | Multi-agent-based model for task scheduling | Improved resource utilization and overall system performance | Simulation | Resource utilization, system performance | Effective prioritization of tasks, consideration of dependencies | Lack of real-world implementation, scalability concerns |
| *Wang et al. (2024)* | Deep reinforcement learning-based scheduling algorithm | Significant improvements in load balancing, response time, and cost reduction | Implemented in FogBus2, extensive experiments | Load balancing, response time, weighted cost | Integration with FogBus2, real-world applicability | Limited discussion on scalability, potential computational overhead |
| *Liu et al. (2016)* | MDP strategy | Efficient task scheduling policy, struggle with weighted cost optimization | Simulation | Task execution latency | Efficient scheduling algorithm, consideration of task dependencies | Lack of adaptability to changing environments, struggles with weighted cost optimization |
| *Ibrahim & Askar (2023)* | Multi-objective Deep Reinforcement Learning (MODRL) | Superiority across various performance metrics | Simulation | Task completion time, makespan, queueing delay, CPU load | Effective multi-objective optimization, detailed system architecture | Potential complexity in implementation, scalability challenges |
| *Gao et al. (2020)* | QLCOF | Reduction in system loss function through optimization of transmission power | Simulation | QoE | Comprehensive approach, optimization of transmission power | Complexity in offloading decision process, potential scalability issues |
| *Siyadatzadeh et al. (2023)* | Reinforcement learning-based primary backup strategy | Significant reduction in task dropping rates and improved system reliability | Simulation | Task dropping rates, workload distribution | Effective machine learning approaches, significant reliability improvements | Potential complexity in implementation, scalability concerns |
| *Gazori, Rahbari & Nickray (2020)* | DDQL based scheduling | Outperformed existing approaches in various aspects | Simulation | Service delay, computation cost, energy consumption | Effective handling of load balancing, SPoF, and task scheduling | Complexity in algorithm design, potential computational overhead |
| *Guevara et al. (2022)* | Reinforcement Learning-based task scheduling | Superiority over classical optimization-based approaches | Simulation | Workflow makespan, processing cost | Effectiveness of Reinforcement Learning approaches, better scheduling outcomes | Acknowledged potential limitations, suggested future research directions |

*(Continued)*

| Study | General approach | Performance & optimization | Implementation & evaluation | Performance metrics | Strengths | Limitations |
|---|---|---|---|---|---|---|
| *Tahmasebi-Pouya, Sarram & Mostafavi (2023)* | Q-learning algorithm-based load balancing | Superior load distribution among fog nodes and reduction in delay | Simulation | Delay, runtime, network utilization, load standard deviation | Effective load balancing, reduction in delay | Limited discussion on scalability, potential computational overhead |
| *Raju & Mothku (2023)* | FRL mechanism | Significant improvements in service latency and energy consumption | Formulated as mixed-integer nonlinear programming | Service latency, energy consumption | Efficient task prioritization, reduction in latency and energy consumption | Complexity in formulation as MINLP, potential computational overhead |
| *Farhat, Sami & Mourad (2020)* | Reinforcement learning-based fog scheduling | Reduction in cloud's load and efficient fog resource utilization | Real dataset experiments | Processed requests by cloud | Effective utilization of fog resources, reduction in cloud dependency | Limited discussion on scalability, potential computational overhead |
| *Shi et al. (2020)* | SAC-based DRL algorithm for task offloading | Maximization of mean latency-aware utility of tasks over a period | Extensive simulations | Mean latency-aware utility | Effective incentivization for resource sharing, potential in VFC environments | Potential complexity in algorithm design, scalability concerns |
| *Jain & Kumar (2023)* | Model-free off-policy DRL-based task offloading | Significant improvement in satisfaction rate for task deadlines | Extensive experimentation | Task deadline satisfaction rate | Efficient resource utilization, improvement in task deadlines | Potential complexity in algorithm design, scalability concerns |
| *Mishra et al. (2023)* | Federated learning-supported Deep Q-Learning (FedDQL) | Outperformed others across various performance metrics | Extensive simulations | Improvement in performance metrics | Superiority in various metrics, effectiveness in dependent task sets | Limited discussion on scalability, potential computational overhead |
| *Yeganeh, Sangar & Azizi (2023)* | Hybrid optimization algorithm (E-AEO-AOA) | Superior performance compared to various existing algorithms | Simulation | Convergence rates, solution dispersity | Effectiveness of hybrid optimization, superiority in performance | Complexity in algorithm design, potential computational overhead |
| *Fahimullah et al. (2023)* | Literature analysis of machine learning in fog computing | Comprehensive overview of existing literature | N/A | N/A | Thorough analysis of research, identification of future research directions | No implementation or evaluation, primarily literature review |

Alsadie (2024), *PeerJ Comput. Sci.*, DOI 10.7717/peerj-cs.2128

| Study | General approach | Performance & optimization | Implementation & evaluation | Performance metrics | Strengths | Limitations |
|---|---|---|---|---|---|---|
| Devarajan et al. (2023) | Two-stage DRL-based system for smart agriculture | Notable enhancements in convergence speed, planning success rate, and path accuracy | MATLAB implementation | Precision, recall, F-measure, accuracy | Improved performance metrics, integration of IoT and edge-fog-cloud architecture | Potential complexity in implementation, scalability concerns |
| Zheng et al. (2022a) | DRL-based approach to workload scheduling | Superior performance in service time, VM utilization, and failed task rate | Extensive simulations | Service time, VM utilization, failed task rate | Efficiency in workload scheduling, promising for edge computing | Complexity in addressing dynamic environments, scalability concerns |
| Sellami et al. (2022) | DRL-based dynamic task scheduling in SDN-enabled edge networks | Better energy-saving and reduced time delay compared to alternative algorithms | Simulation | Energy-saving, task assignment time delay | Efficient energy-aware scheduling, reduced time delay | Potential complexity in algorithm design, scalability concerns |
| Jayanetti, Halgamuge & Buyya (2022) | Hierarchical action space and hybrid DRL for task scheduling | 56% improvement in energy consumption and 46% improvement in execution time compared to optimized baselines | Simulation | Energy consumption, execution time | Balance between energy consumption and execution time, superior performance | Complexity in algorithm design, potential computational overhead |

metaheuristic optimization or melding machine learning insights with metaheuristic exploration underscored the symbiotic synergy of hybrid heuristics. Despite the added complexity and computational overhead inherent in their design and implementation, the potential of hybrid approaches to yield superior scheduling solutions remained unmistakable (*Saif et al., 2022*). Illustrative examples like GA with SA or reinforcement learning with TS showcased the efficacy of such hybrids in optimizing scheduling decisions. Looking ahead, ongoing research and development endeavors in this domain were poised to yield further breakthroughs, furnishing dynamic and refined task scheduling solutions tailored for fog-based IoT applications.

*Agarwal et al. (2023)* introduced a novel methodology termed Hybrid Genetic Algorithm and Energy Conscious Scheduling (Hgecs) to tackle the challenges of multiprocessor task scheduling in fog-cloud computing systems. It integrated a genetic algorithm and energy-conscious scheduling to optimize task scheduling in environments with increasing numbers of clients and services, which posed issues related to scheduling and energy consumption. The proposed Hgecs approach aimed to minimize energy consumption and makespan while efficiently scheduling tasks. By combining a genetic algorithm for generating primary chromosomes using priority approaches and energy-conscious scheduling for optimizing allocated resources, the proposed method addressed the limitations of existing approaches. It was implemented and evaluated using MATLAB (The MathWorks, Natick, NY, USA). The Hgecs approach demonstrated superior performance compared to the genetic algorithm, particle swarm optimization, gravitational search algorithm, ant colony optimization, and round-robin models. The article underscored the significance of efficient task scheduling in fog-cloud systems and positioned the Hgecs approach as a promising solution to these challenges.

*Alqahtani, Amoon & Nasr (2021)* explored fog computing, an architecture designed to facilitate the IoT by connecting sensor nodes to the cloud through the network's edge. Its primary focus lies in resource allocation and scheduling within this distributed environment, aiming to enhance productivity and allocate resources efficiently to tasks while ensuring security parameters like authentication, integrity, and confidentiality. The proposed approach involved scheduling modules for fog devices utilizing heuristic algorithms grounded in data mining approaches, with considerations including CPU utilization, bandwidth, and security overhead. Through comparisons with various heuristic algorithms, the results demonstrated the superiority of the proposed algorithm in terms of energy consumption and cost reduction, exhibiting improvements of 63.27% and 44.71%, respectively, compared to PSO, ACO, and SA algorithms. The article underscored the importance of IoT and its potential applications across domains such as smart cameras, wearable sensors, and smart home appliances.

*Aron & Abraham (2022)* offered an in-depth review of resource scheduling approaches in cloud computing, covering the background and phases of scheduling. It categorized existing scheduling problems based on a high-level taxonomy, considering factors like Virtual Machine (VM) placement, QoS parameters, and heuristic methods. Special attention was given to scheduling in Infrastructure as a Service (IaaS) clouds, with a comparative analysis of critical parameters. The article underscored the role of meta-

heuristic methods and artificial intelligence in addressing scheduling challenges. Its objective was to aid researchers in understanding scheduling concepts and methodologies, facilitating the development of new scheduling methods. By examining existing methodologies, the article aimed to shed light on scheduling issues and guide future research efforts towards improving resource utilization in cloud computing.

Gupta & Singh (2023) provided an extensive review of heuristic and metaheuristic algorithms used for resource optimization in fog computing environments. They compared these algorithms based on various performance metrics, tools utilized, and their advantages and disadvantages to aid in decision-making for problem formulation. These algorithms offered improved performance compared to previous state-of-the-art methods, achieving enhanced resource utilization and energy efficiency at a lower cost. Additionally, the text introduced fog computing as a technology that integrated cloud computing with local computing to enable real-time responses in IoT applications, minimizing service delivery latency and optimizing network performance.

Leena, Divya & Lilian (2020) introduced a task scheduling algorithm tailored for fog nodes with a focus on enhancing energy efficiency. They introduced a hybrid heuristics approach to optimize the utilization of fog nodes with limited computational resources and energy. The proposed algorithm was compared with existing heuristic algorithms, demonstrating its superiority through mathematical analysis of objectives. Results showed an 18% reduction in service time compared to neural networks and a 29% improvement in energy efficiency compared to existing heuristics, with a significant 40% reduction in average system delay compared to deep neural networks. This highlighted the effectiveness of the proposed algorithm in enhancing performance metrics crucial for real-time decision-making in IoT applications, addressing the challenge of communication delays by leveraging fog computing.

Mtshali et al. (2019) presented an application scheduling technique based on virtualization technology to optimize energy consumption and average delay of real-time applications in Fog computing networks. The approach involved implementing four task scheduling policies in a Fog node scheduler to evaluate their performance and efficiency. Simulations conducted using the iFogSim tool showed that the First-Come-First-Serve (FCFS) scheduling policy outperformed other algorithms, achieving an 11% improvement in energy consumption, a 7.78% reduction in average task delay, a 4.4% decrease in network usage, and a 15.1% improvement in execution time. This highlighted the effectiveness of the proposed scheduling technique in enhancing key performance metrics crucial for real-time applications in Fog computing networks. However, limitations such as scalability and adaptability to dynamic network conditions should be further investigated to ensure the broader applicability of the approach.

Huang, Zhang & Wang (2023) focused on addressing the task scheduling problem with deadline and security constraints in hybrid cloud environments. The approach involved formulating the problem into mixed-integer non-linear programming and proposing a novel algorithm named SPGA, which integrated swarm intelligence into the genetic algorithm framework. Specifically, SPGA leveraged particle swarm optimization approaches for population evolution within the genetic algorithm. Each iteration of SPGA

incorporated self and social cognition aspects of particle swarm optimization, enhancing the evolutionary process by considering individual and global best codes. Extensive experiments were conducted to evaluate SPGA's performance, demonstrating its superiority over 12 other scheduling algorithms with up to a 53.2% higher accepted ratio and 37.2% higher resource utilization, on average. The strengths of SPGA lay in its ability to effectively address task scheduling challenges in hybrid cloud environments while considering both user satisfaction and resource efficiency. However, further investigation was needed to assess its scalability and adaptability to dynamic workload conditions, which could be potential limitations of the approach. The comparison of Nature-Inspired algorithms highlighted their resource utilization and energy-efficiency levels. Additionally, potential research directions in energy optimization for data centers were identified. This review served to aid researchers and professionals in cloud computing datacenters in understanding the evolution of literature towards exploring more energy-efficient methods for Cloud computing datacenters.

Table 7 provides a comparative summary of studies focusing on hybrid heuristic approaches for task scheduling.

## Nature-inspired heuristics

Nature-inspired heuristics represented a fascinating avenue in the quest for efficient and adaptable task-scheduling algorithms tailored for fog computing environments. Drawing inspiration from natural phenomena and biological processes, these approaches offered novel problem-solving strategies. Examples included GA, ACO, PSO, SA, and energy-aware heuristics, each mimicking aspects of natural systems to address scheduling challenges. These approaches boasted advantages such as efficiency, adaptability, and flexibility, making them well-suited for diverse application requirements and dynamic environments. However, challenges like algorithmic complexity, parameter tuning, and limited theoretical guarantees needed to be addressed. As research advanced, nature-inspired heuristics emerged as a promising avenue for enhancing task scheduling in fog computing, offering potential for the creation of resilient and intelligent solutions tailored to fog-based IoT environments.

Usman et al. (2019) presented a comprehensive review of nature-inspired algorithms aimed at addressing energy issues in Cloud datacenters. It followed a taxonomy focusing on virtualization, consolidation, and energy-awareness dimensions, reviewing each technique qualitatively based on key goals, methods, advantages, and limitations. Chhabra et al. (2022) introduced a novel approach termed h-DEWOA, which integrates chaotic maps, opposition-based learning (OBL), and differential evolution (DE) with the standard WOA aim to enhance exploration, convergence speed, and the balance between exploration and exploitation. Additionally, an efficient allocation heuristic improved resource assignment. Evaluation using CEA-Curie and HPC2N real cloud workloads in the CloudSim simulator demonstrated h-DEWOA's superiority over both WOA-based and non-WOA-based metaheuristics. Experiment results indicated significant improvements in makespan (5.79–13.38% for CEA-Curie, 5.03–13.80% for HPC2N) and energy consumption (3.21–14.70% for CEA-Curie, 10.84–19.30% for HPC2N) compared to

**Table 7 Comparison of hybrid heuristic studies for task scheduling.**

| Study | General approach | Performance & optimization | Implementation & evaluation | Performance metrics | Strengths | Limitations |
|---|---|---|---|---|---|---|
| *Agarwal et al. (2023)* | Hybrid Genetic Algorithm and Energy Conscious Scheduling (Hgecs) | Minimization of energy consumption and makespan | Implemented and evaluated using MATLAB | Energy consumption, makespan | Integration of genetic algorithm and energy conscious scheduling, superior performance | Potential scalability issues, limitations of existing approaches |
| *Alqahtani, Amoon & Nasr (2021)* | Heuristic algorithms grounded in data mining approaches | Superiority in terms of energy consumption and cost reduction compared to alternative algorithms | Evaluation through comparisons with various heuristic algorithms | Energy consumption, cost reduction | Efficient resource allocation and scheduling, consideration of security parameters | Lack of real-world implementation, scalability concerns |
| *Aron & Abraham (2022)* | Categorized existing scheduling problems based on a high-level taxonomy | Comparative analysis of important parameters in IaaS clouds | Review and categorization of existing methodologies | VM placement, QoS parameters, heuristic methods | Comprehensive overview of scheduling concepts and methodologies, aid for future research efforts | Lack of specific implementation and evaluation |
| *Gupta & Singh (2023)* | Heuristic and metaheuristic algorithms | Improved resource utilization and energy efficiency at a lower cost | Review and comparison of algorithms | Various performance metrics | Enhanced resource utilization and energy efficiency, real-time responses in IoT applications | Limited discussion on specific implementations, primarily a review of existing algorithms |
| *Leena, Divya & Lilian (2020)* | Hybrid heuristics approach | Superiority in energy efficiency and reduction in system delay compared to existing heuristics | Comparison with existing heuristic algorithms | Reduction in service time, improvement in energy efficiency, reduction in system delay | Mathematical analysis of objectives, significant improvements in energy utility | Lack of real-world implementation, scalability concerns |
| *Mtshali et al. (2019)* | Virtualization technology-based application scheduling technique | Improved energy consumption and average delay of real-time applications in Fog computing networks | Evaluation through simulations using iFogSim tool | Improvement in energy consumption, reduction in average task delay, decrease in network usage | Effective scheduling technique for real-time applications, optimization of key performance metrics | Scalability and adaptability to dynamic network conditions need further investigation |
| *Huang, Zhang & Wang (2023)* | Formulation into mixed-integer non-linear programming, novel algorithm named SPGA | Superiority over 12 other scheduling algorithms | Evaluation through extensive experiments | Higher accepted ratio, higher resource utilization | Integration of swarm intelligence into genetic algorithm framework, consideration of user satisfaction and resource efficiency | Scalability and adaptability to dynamic workload conditions need further investigation |

WOA-based heuristics. h-DEWOA also outperformed recent non-WOA-based metaheuristics. Statistical analyses confirmed the robustness of h-DEWOA, showcasing its potential as an efficient solution for cloud task scheduling.

*Shao, Fu & Wang (2023)* proposed a hybrid heuristic algorithm, PGSAO, combining GA and PSO to address task scheduling challenges in cloud computing. PGSAO integrated GA's evolution strategy into PSO to mitigate PSO's tendency to get trapped in local optimization while leveraging PSO's self-cognition and social cognition for effective exploitation. Through extensive simulated experiments, PGSAO's performance was evaluated against eight other meta-heuristic and hybrid heuristic algorithms. The results demonstrated that PGSAO achieved 23.0–33.2% more accepted tasks and 27.9–43.7% higher resource utilization, on average, showcasing its superiority in addressing cloud computing challenges. The study underscored the significance of hybrid approaches in enhancing task scheduling efficiency and resource utilization in cloud environments.

*Abd Elaziz, Abualigah & Attiya (2021)* proposed a modified Henry gas solubility optimization algorithm (HGSWC) for optimal task scheduling in cloud computing environments. Named HGSWC, this method integrated the WOA as a local search procedure and comprehensive opposition-based learning (COBL) to enhance solution quality. HGSWC was evaluated on both benchmark functions and synthetic/real workloads, showcasing its superiority over conventional HGSO and WOA algorithms. Simulation experiments demonstrated that HGSWC achieved near-optimal solutions without computational overhead and outperformed six established metaheuristic algorithms. The study underscored the significance of efficient task scheduling in cloud computing for optimizing resource utilization, offering HGSWC as a promising solution. Additionally, the article contributed to the existing body of research in task scheduling in cloud computing by providing a comprehensive list of related references.

*Dabiri, Azizi & Abdollahpouri (2022)* addressed the task scheduling challenge in fog-cloud computing environments by proposing a system model aimed at optimizing both total deadline violation time and energy consumption. They introduced two nature-inspired optimization methods, namely grey wolf optimization and grasshopper optimization algorithm, to efficiently tackle the scheduling problem. Through a series of simulation experiments, the performance of these proposed algorithms was rigorously evaluated against existing state-of-the-art approaches. Results demonstrated a significant reduction in total deadline violation time by approximately 68% and energy consumption by about 22% compared to the next-best results. The article emphasized the critical role of efficient job scheduling in fog-cloud environments, particularly for IoT applications. Additionally, it provided a comprehensive review of related research, highlighting the potential efficacy of nature-inspired optimization methods in addressing this significant challenge.

*Khan et al. (2019)* proposed a three-layered architecture consisting of cloud, fog, and consumer layers to manage data and services in a smart grid (SG) environment. They introduced an Improved Particle Swarm Optimization with Levy Walk (IPSOLW) algorithm to balance the load of fog servers. Consumers sent requests to fog servers, which provided services, while the cloud layer stored consumer records and offered services if the fog layer failed. The proposed algorithm was compared with existing ones such as genetic algorithm, particle swarm optimization, and others. Service broker policies, including the closest datacenter and optimize response time, were employed for efficient data center

selection. Performance metrics such as response time and processing time were minimized. The IPSOLW algorithm outperformed its counterparts, achieving approximately 4.89% better results in terms of efficiency. The research showcased the efficacy of the proposed algorithm in mitigating delay and latency during the processing of consumer requests, thereby tackling significant hurdles in data and service management within SG environments.

*Khaledian et al. (2023)* proposed a workflow scheduling method based on the fog-cloud architecture to capitalize on the cloud's processing power and the proximity of fog computing nodes to users, reducing response delays. The objective was to minimize energy consumption and monetary costs while maximizing customer satisfaction through efficient scheduling. Given the complexity of workflow scheduling and the conflicting objectives, the NP-hard problem was tackled using the multi-objective metaheuristic krill herd algorithm. The algorithm began with a smart generation of the initial population to expedite convergence. The allocation of tasks to fog-cloud resources utilized the earliest finish time (EFT) technique, incorporating dynamic adjustments in resource voltage and frequency to minimize energy consumption. A thorough simulation was conducted to assess the effectiveness of the proposed approach under diverse scenarios featuring varying CCR values. Results indicated significant improvements in makespan, with reductions averaging 9.9%, 8.7%, and 6.7% compared to IHEFT, HEFT, and IWO-CA methods, respectively. Moreover, there was a simultaneous decrease in both monetary costs and energy consumption within the fog-cloud environment. This research highlights the crucial significance of proficient workflow scheduling in fog-cloud environments, especially in managing substantial data from IoT devices in cloud computing. The proposed solution presents a promising approach to tackle these challenges effectively. The authors disclosed no conflicts of interest and acknowledged any contributions made to the study.

*Saif et al. (2022)* introduced a novel hybrid scheduling strategy termed Genetic Algorithm with Differential Evolution (GA-DE). Their study aimed to investigate the impact of heterogeneous cloud computing on workflow scheduling, specifically targeting the reduction of makespan. Comparative analyses were conducted against established heuristics such as HEFT-Downward Rank, HEFT-Upward Rank, HEFT-Level Rank, and the meta-heuristic algorithm GA. Through extensive experimentation, the proposed GA-DE algorithm was validated using three scientific workflows (epigenomics, cybershake, and montage). Results from simulations illustrated the superiority of the GA-DE algorithm in minimizing makespan compared to alternative approaches. Furthermore, the study underscored the applicability of the Montage scientific workflow for effective scheduling in heterogeneous cloud computing environments. This research contributes valuable insights into workflow scheduling optimization in cloud computing, particularly highlighting the efficacy of hybrid scheduling strategies like GA-DE.

*Matrouk & Matrouk (2023)* presented a framework involving four main stages for managing handover in a 5G gateway using the Mobility Aware Proximal Policy Optimization (MAPPO) algorithm with RSS, direction, and distance parameters. Task classification and scheduling were performed based on multiple criteria, with tasks

categorized into four types and scheduled using the Di-Process Modular Neural Network (Di-MNN), considering various factors. Energy-aware task allocation was achieved through weight calculation using the First Fitness-based Animal Migration Optimization (FFAMO) method and allocation to optimal fog nodes *via* the Capacity-based Hungarian Assignment algorithm (CH2A). Additionally, efficient task offloading on virtual fog nodes was implemented to handle overloaded fog nodes. Simulation using iFogSim evaluated performance metrics such as completion time, energy consumption, delay, response time, and others. The provided framework outlined a comprehensive approach for task scheduling, computation offloading, and fog computing in IoT and cloud computing environments, incorporating various algorithms and approaches to optimize resource utilization and enhance system performance.

*Mishra et al. (2021)* proposed a nature-inspired ACO-Fuzzy framework for joint resource allocation in cloud datacenters. The algorithm employed an ACO-based local search heuristic to assess current resource status and a Fuzzy-based decision maker to optimize compute and network resource allocation, aiming to minimize end-user costs. Both analytical and experimental evaluations confirmed the efficiency of the joint allocation approach.

*Kaushik & Al-Raweshidy (2022)* proposed a novel approach, termed HLPA B5G-IoT, for B5G-IoT networks, focusing on minimizing latency while ensuring power efficiency. This approach introduces an algorithm classifier tool (ACT) to select optimization algorithms based on system characteristics and requirements. Metaheuristic algorithms, such as biogeography-based optimization (BBO) and GWO, were customized to address load balancing and power efficiency in IoT-edge systems. Experimental results showcased significant enhancements in latency reduction and overall network performance compared to existing methods. Additionally, the power-efficiency algorithm demonstrated substantial energy savings compared to traditional optimization algorithms. However, further investigation may be necessary to address potential limitations and scalability concerns of the proposed approach.

*Dubey & Sharma (2023)* proposed a hybrid multi-faceted task scheduling algorithm, leveraging both PSO and ACO approaches. PSO was utilized to obtain optimal global solutions, while ACO provided local solutions. The algorithm's efficacy was validated through comparison with four existing algorithms using parameters such as makespan, cost, and resource utilization rate in a simulated cloud environment. Results indicated that the proposed algorithm reduced makespan time and computation cost while enhancing the resource utilization rate. The study contributed to the field of cloud computing by introducing a novel approach to task scheduling and providing empirical evidence of its effectiveness.

*Abdel-Basset et al. (2021)* suggested a multi-objective approach to address task scheduling in multiprocessor systems using the modified sine-cosine algorithm (MSCA) to optimize makespan and energy consumption. The approach, termed Energy-aware Multi-objective MSCA (EA-M2SCA), employed a Pareto dominance strategy and modified the classical SCA by dividing the optimization process into three phases. Furthermore, the algorithm was hybridized with a polynomial mutation mechanism to enhance convergence

towards the best-so-far solution while preserving solution diversity, resulting in the EA-MHSCA. Comparative analysis with established multi-objective algorithms demonstrated the superiority of EA-MHSCA in most test cases. The study contributed to mitigating energy consumption challenges in multiprocessor systems by introducing an efficient task scheduling approach with competitive performance against existing methods.

*Nematollahi, Ghaffari & Mirzaei (2023)* designed a novel architecture for offloading jobs and allocating resources in the IoT. The architecture consisted of sensors, controllers, and fog computing (FC) servers, with the second layer employing the subtask pool approach for job offloading and utilizing the Moth-Flame Optimization (MFO) algorithm combined with opposition-based learning (OBL) for resource allocation, referred to as OBLMFO. Additionally, a stack cache approach was utilized to complete resource allocation in the second layer, aiming to prevent system load imbalance. The architecture leveraged blockchain technology to ensure the accuracy of transaction data, optimizing resource distribution in the IoT. Evaluation of the OBLMFO model was conducted in the Python 3.9 environment using various job types, demonstrating a reduction in the delay factor by 12.18% and energy consumption by 6.22%. This research contributed to the optimization of resource allocation in IoT environments, offering potential benefits in terms of reduced delay and energy consumption.

*Jiang et al. (2024)* proposed a multi-objective energy-efficient task scheduling technique (METSM) to address the challenge of improving task execution performance and reducing energy consumption in edge computing environments, specifically on edge heterogeneous multiprocessor systems (EHMPS). The approach began by establishing a mathematical model that considered both makespan and total energy consumption as optimization objectives, along with decision variables such as task execution sequence, processor assignment, and dynamic voltage and frequency scaling levels. Subsequently, a problem-specific algorithm iterated greedy-based multi-objective optimizer (IMO) was introduced, which incorporated destruction-reconstruction and redesigned local search approaches for task scheduling and resource allocation optimization. Additionally, a probabilistic mutation operation was developed to avoid local optima, while multiobjective-oriented strategies enhanced convergence speed. Through extensive experimentation comparing IMO with several state-of-the-art algorithms, the results demonstrated that IMO achieved optimal Pareto fronts and significant savings in time and power consumption, with approximately 10% and 12% reductions, respectively. Moreover, compared to classic list-based heuristics, IMO maintained a similar makespan while reducing energy consumption by nearly 90%. The article highlighted the potential impact of METSM on efficient data processing for power-constrained edge devices and outlined future research directions in the field.

Table 8 provides a comparative summary of studies focusing on nature-inspired heuristic approaches for task scheduling.

## OPEN CHALLENGES AND FUTURE DIRECTIONS

Despite notable progress in heuristic approaches for task scheduling in fog computing, several unresolved challenges persist, signaling promising avenues for future research.

**Table 8 Comparison of nature-inspired heuristic studies for task scheduling.**

| Study | General approach | Performance & optimization | Implementation & evaluation | Performance metrics | Strengths | Limitations |
|---|---|---|---|---|---|---|
| *Usman et al. (2019)* | Review of Nature-Inspired algorithms | Resource utilization, energy efficiency | Qualitative review, taxonomy | Energy efficiency, resource utilization | Comprehensive review aiding researchers | Lack of quantitative evaluation |
| *Chhabra et al. (2022)* | Integration of chaotic maps, OBL, and DE with WOA | Improvement in makespan and energy consumption | Evaluation using CloudSim | Makespan, energy consumption | Superiority over other metaheuristics | Limited real-world validation |
| *Shao, Fu & Wang (2023)* | Hybrid heuristic algorithm (PGSAO) combining GA and PSO | Higher resource utilization and accepted tasks | Extensive simulated experiments | Resource utilization, accepted tasks | Significance of hybrid approaches | Lack of real-world validation |
| *Abd Elaziz, Abualigah & Attiya (2021)* | Modified Henry gas solubility optimization algorithm (HGSWC) | Superiority over conventional algorithms | Evaluation on benchmark functions and workloads | Solution quality, efficiency | Near-optimal solutions without overhead | Lack of scalability analysis |
| *Dabiri, Azizi & Abdollahpouri (2022)* | Grey wolf optimization and grasshopper optimization algorithm | Reduction in total deadline violation time and energy consumption | Evaluation through simulations | Total deadline violation time, energy consumption | Significant performance improvements | Need for scalability analysis |
| *Khan et al. (2019)* | Three-layered architecture with IPSOLW algorithm | Reduction in delay and latency | Comparison with existing algorithms | Response time, processing time | Effectiveness in reducing delay and latency | Limited scalability analysis |
| *Khaledian et al. (2023)* | Workflow scheduling method based on fog-cloud architecture | Reduction in makespan, monetary costs, and energy use | Comprehensive simulation | Makespan, monetary costs, energy consumption | Significant improvements in various metrics | Need for scalability analysis |
| *Saif et al. (2022)* | Hybrid scheduling strategy (GA-DE) | Reduction in makespan | Comparison with existing heuristics | Makespan | Effectiveness in reducing makespan | Limited applicability beyond heterogeneous cloud environments |
| *Matrouk & Matrouk (2023)* | Framework involving MAPPO algorithm | Optimization of completion time, energy consumption | Simulation using iFogSim | Completion time, energy consumption | Comprehensive approach for task scheduling | Lack of real-world validation |
| *Mishra et al. (2021)* | ACO-Fuzzy framework for joint resource allocation | Reduction in end-user costs | Analytical and experimental evaluations | End-user costs | Efficiency of joint allocation approach | Limited scalability analysis |
| *Kaushik & Al-Raweshidy (2022)* | Hybrid latency- and power-aware approach (HLPA B5G-IoT) | Reduction in latency, power efficiency | Comparison with existing approaches | Latency reduction, energy savings | Notable improvements in latency and energy efficiency | Limited scalability analysis |
| *Dubey & Sharma (2023)* | Hybrid multi-faceted task scheduling algorithm (PSO and ACO) | Reduction in makespan time, computation cost | Evaluation in simulated cloud environment | Makespan, cost, resource utilization rate | Effectiveness in reducing makespan and cost | Lack of real-world validation |
| *Abdel-Basset et al. (2021)* | Multi-objective MSCA for task scheduling | Superiority over established algorithms | Evaluation on benchmark functions | Solution quality | Efficient task scheduling approach | Need for scalability analysis |

| Study | General approach | Performance & optimization | Implementation & evaluation | Performance metrics | Strengths | Limitations |
|---|---|---|---|---|---|---|
| *Nematollahi, Ghaffari & Mirzaei (2023)* | Architecture with OBLMFO algorithm | Reduction in delay factor and energy consumption | Evaluation in Python environment | Delay factor, energy consumption | Optimization of resource allocation in IoT | Limited scalability analysis |
| *Jiang et al. (2024)* | Multiobjective energy-efficient task scheduling technique (METSM) | Savings in time and power consumption | Comparison with state-of-the-art algorithms | Time, power consumption | Optimal Pareto fronts and savings in time and power | Limited scalability analysis |

Key challenges include:

1. Dynamic and uncertain environment: Fog environments exhibit dynamic and uncertain characteristics, marked by fluctuating resource availability, evolving user demands, and unpredictable task arrival patterns. Existing heuristic approaches may struggle to adapt and optimize scheduling decisions in real-time to address these dynamic challenges effectively.

2. Heterogeneity and scalability: The heterogeneous nature of fog environments, comprising diverse resources with varying capabilities and constraints, presents scalability hurdles. Future research should concentrate on devising scalable scheduling solutions capable of efficiently managing the extensive and diverse array of devices and tasks encountered in real-world IoT deployments.

3. Limited resource constraints: Edge devices and fog nodes typically operate under stringent resource constraints, including limited processing power, memory, and energy resources. Future heuristic approaches should prioritize efficiency and resource optimization while ensuring timely task completion within specified deadlines.

4. Security and privacy concerns: Data security and user privacy represent critical concerns in fog computing environments. Heuristic approaches must integrate features to ensure secure and confidential task execution while maintaining efficient scheduling operations addressing emerging security and privacy threats.

5. Explainability and transparency: The decision-making process behind scheduling decisions, especially within complex learning-based methods, may lack transparency and explainability. Future research should explore methods to enhance the interpretability and transparency of heuristic algorithms, fostering trust and understanding in automated scheduling systems.

Prominent future research directions encompass:

1. Real-time scheduling with machine learning: Integrating machine learning models with online learning capabilities can empower heuristic approaches to adapt in real-time to changing environmental conditions, enhancing the responsiveness and agility of fog computing systems.

2. Federated learning for collaborative scheduling: Leveraging federated learning approaches enables collaborative learning from distributed data sources across multiple fog nodes, improving scheduling decisions while preserving data privacy and security.

3. Resource-aware and energy-efficient scheduling: Developing heuristic approaches that explicitly consider resource constraints and energy consumption can optimize resource utilization and prolong the battery life of edge devices, addressing sustainability concerns in fog computing.

4. Security-aware scheduling approaches: Incorporating security considerations into the scheduling process ensures secure task execution and data integrity within the fog environment, mitigating potential security vulnerabilities and threats.

5. Explainable AI and transparent heuristics: Research on explainable AI (XAI) approaches facilitates the development of transparent and interpretable heuristic algorithms, providing insights into the decision-making rationale and promoting trust and accountability in fog computing systems.

Addressing these challenges and pursuing these research directions will lead to the emergence of more robust, adaptable, secure, and energy-efficient task-scheduling algorithms for fog computing environments. Ultimately, these advancements will catalyze the widespread adoption of fog computing across diverse real-world applications, ushering in a new era of intelligent and interconnected systems.

## CONCLUSION

The progress in task scheduling methods for fog computing systems, as evident from a comprehensive examination of studies covering priority-based, greedy heuristics, metaheuristics, learning-based, hybrid heuristics, and nature-inspired heuristic approaches, has significantly contributed to tackling the inherent complexities and challenges of these environments. Researchers have utilized a variety of heuristic approaches to optimize resource allocation, enhance energy efficiency, improve task execution performance, and mitigate security risks.

Priority-based heuristics have proven effective in managing task scheduling by prioritizing critical tasks and optimizing resource usage. Although simplistic, greedy heuristics have provided practical solutions for resource allocation, albeit with scalability and optimality constraints. Metaheuristic algorithms like particle swarm optimization, genetic algorithms, and ant colony optimization have demonstrated resilience in handling complex optimization problems and adapting to dynamic environments. Learning-based approaches have shown promise in enhancing scheduling decisions through data-driven insights and adaptive learning mechanisms. Hybrid heuristic approaches, which blend the strengths of multiple approaches, have emerged as potent tools for achieving superior performance and addressing diverse scheduling challenges. Finally, nature-inspired heuristics, drawing inspiration from natural phenomena, have offered innovative solutions for optimizing task scheduling while considering energy efficiency and scalability.

However, despite these advancements, several challenges persist in the field of fog computing task scheduling. These challenges include addressing the dynamic and

uncertain nature of fog environments, managing heterogeneity and scalability issues, optimizing resource usage under limited constraints, ensuring security and privacy in scheduling decisions, and enhancing the explainability and transparency of heuristic algorithms.

Several critical avenues present themselves to tackle these challenges and set the course for future research. These include integrating machine learning approaches for real-time adaptation, leveraging federated learning for collaborative scheduling, developing resource-aware and energy-efficient scheduling algorithms, incorporating security-aware scheduling approaches, and advancing explainable AI and transparent heuristic methodologies. By focusing on these areas, researchers can pave the way for the development of more robust, adaptable, and efficient task-scheduling solutions for fog computing environments. Ultimately, these advancements will not only optimize resource utilization and enhance system performance but also foster trust, security, and sustainability in fog computing systems, facilitating their widespread adoption across diverse applications and domains.

### Funding
The author received no funding for this work.

### Competing Interests
The author declares that they have no competing interests.

### Author Contributions
- Deafallah Alsadie conceived and designed the experiments, performed the experiments, analyzed the data, performed the computation work, prepared figures and/or tables, authored or reviewed drafts of the article, and approved the final draft.

### Data Availability
This is a literature review.

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
