# Peer review of "Advancements in heuristic task scheduling for IoT applications in fog-cloud computing: challenges and prospects"

_PeerJ Computer Science, doi:10.7717/peerj-cs.2128_

## Round 0.1 · accepted · Accept

Both reviewers consider your paper a valuable contribution to the field. Please note that reviewer 2 brings up some points that could enhance the paper, such as adding practical implementation examples. Considering this is a literature review article, I have accepted the paper as is.

Reviewer 1 ·

Basic reporting

The paper is written in very clear language and has an equally clear and easy to follow structure. Its introduction gives a concise and informative overview of the topic, but may benefit from some additional detail for newcomers. Literature references are concise when appropriate, while naturally extensive in the actual review part. Since I am not from the particular field, I cannot comment on whether specific references might be missing or not.
Regarding structure, the paper begins by clearly defining its methodology, proceeds to give some high-level statistical insights about the ~100 papers reviewed, and gives summaries and comparisons of scheduling approaches taken in numerous representative publications. Finally, indications are given regarding potential future research directions. As a result, the review is easy to follow and accessible.

There is a number of more or less related reviews in the field, including https://doi.org/10.1002/cpe.6432 and https://doi.org/10.1007/s10723-019-09491-1 , some of which have a considerably broader scope than the present work. However, since some of these are relatively old (taking into account a fast-moving field) and others do not appear to cover heuristic task scheduling in comparable detail, I believe the present work is a valuable contribution.
As part of the introduction, references to related reviews should be added, clearly delineating their scopes. In particular, it is important to direct readers interested in a higher-level overview of fog computing to more suitable studies.

Experimental design

The article aligns well with the journal, and the investigation is performed according to established procedures (in particular taking the PRISMA approach) that are clearly stated.
Not being an expert on the field of fog computing, I cannot assess if there may be aspects of the field missing. The presented material, however, gives an objective and neutral survey.
Sources are referenced appropriately, and the review is very well-organized into subsections.

Validity of the findings

The paper has a dedicated section discussing open challenges and possible directions for future research.

Reviewer 2 ·

Basic reporting

The paper presents a systematic review of heuristic task scheduling methods for fog computing systems, targeting IoT applications. It explores various scheduling approaches, including priority-based, greedy, metaheuristic, learning-based, hybrid, and nature-inspired heuristics. The review is structured according to the PRISMA guidelines, ensuring methodological rigor, and it highlights the strengths and limitations of each approach, identifies key challenges in the domain, and suggests future research directions.

Strengths:

Extensive Literature Review: The systematic review is thorough, covering a wide range of heuristic methods. The paper benefits from a structured approach that rigorously assesses the suitability of each method for fog computing environments.

Clear Identification of Research Gaps: By systematically analyzing existing literature, the paper effectively pinpoints the gaps and challenges in the field. This serves as a valuable resource for researchers aiming to further the development of heuristic task scheduling in fog computing.

Future Research Directions: The detailed discussion on future research directions is commendable. Proposals for incorporating advanced machine learning techniques and federated learning into heuristic task scheduling are particularly forward-thinking and aligned with current technological trends.

Weaknesses:

Lack of Empirical Evidence: The review, while comprehensive in its literature scope, lacks empirical studies or practical implementation examples. This limits the paper’s ability to demonstrate the real-world applicability and effectiveness of the reviewed methodologies.

Limited Discussion on Real-Time Processing: Given the dynamic nature of IoT environments, the paper could strengthen its impact by addressing the challenges and solutions related to real-time data processing and task scheduling.

Superficial Treatment of Security Issues: Security concerns are crucial in IoT and fog computing; however, they receive minimal attention in this review. A more in-depth analysis of security-aware scheduling techniques would be beneficial.

Experimental design

The systematic literature review process is briefly explained in paper.

Validity of the findings

No comments.